# Amphiphilic nanocarrier-induced modulation of PLK1 and miR-34a leads to improved therapeutic response in pancreatic cancer

Hadas Gibori[1], Shay Eliyahu[1], Adva Krivitsky[1], Dikla Ben-Shushan[1], Yana Epshtein[1], Galia Tiram[1], Rachel Blau[1], Paula Ofek[1], Joo Sang Lee[2], Eytan Ruppin[1,2,3], Limor Landsman [4], Iris Barshack[5,6], Talia Golan[5], Emmanuelle Merquiol[7], Galia Blum[7] & Ronit Satchi-Fainaro [1]

The heterogeneity of pancreatic ductal adenocarcinoma (PDAC) suggests that successful treatment might rely on simultaneous targeting of multiple genes, which can be achieved by RNA interference-based therapeutic strategies. Here we show a potent combination of microRNA and siRNA delivered by an efficient nanocarrier to PDAC tumors. Using proteomic-microRNA profiles and survival data of PDAC patients from TCGA, we found a novel signature for prolonged survival. Accordingly, we used a microRNA-mimic to increase miR-34a together with siRNA to silence PLK1 oncogene. For in vivo dual-targeting of this combination, we developed a biodegradable amphiphilic polyglutamate amine polymeric nanocarrier (APA). APA-miRNA–siRNA polyplexes systemically administered to orthotopically inoculated PDAC-bearing mice showed no toxicity and accumulated at the tumor, resulting in an enhanced antitumor effect due to inhibition of MYC oncogene, a common target of both miR-34a and PLK1. Taken together, our findings warrant this unique combined polyplex's potential as a novel nanotherapeutic for PDAC.

---

[1] Department of Physiology and Pharmacology, Sackler Faculty of Medicine, Tel Aviv University, Tel Aviv 69978, Israel. [2] Department of Computer Science and Center for Bioinformatics and Computational Biology, University of Maryland, College Park, MD 20742, USA. [3] Blavatnik School of Computer Sciences, Tel Aviv University, Tel Aviv 69978, Israel. [4] Department of Cell and Developmental Biology, Sackler Faculty of Medicine, Tel Aviv University, Tel Aviv 69978, Israel. [5] Department of Pathology, Sheba Medical Center, Tel Hashomer 52621, Israel. [6] Department of Pathology, Sackler Faculty of Medicine, Tel Aviv University, Tel Aviv 69978, Israel. [7] The Institute for Drug Research, School of Pharmacy, Faculty of Medicine, Ein Kerem Campus, The Hebrew University, Jerusalem, Israel. Correspondence and requests for materials should be addressed to R.S-F. (email: ronitsf@post.tau.ac.il)

Despite the better understanding of pancreatic ductal adenocarcinoma (PDAC) molecular biology in the past decade, almost all targeted therapies have failed to demonstrate efficacy in late phase clinical trials[1]. A promising strategy to treat cancer is knocking-down the expression of specific cancer-promoting genes by RNA interference (RNAi)-based therapeutics, such as small interfering RNA (siRNA) and microRNA (miRNA)[2]. siRNAs are currently under investigation in several clinical trials for cancer treatment[3]. As opposed to siRNAs, which target a specific gene, miRNAs regulate hundreds of mRNA targets at once, thus making them an even more attractive tool to treat cancer[4]. miRNAs have been shown to be dysregulated in various human cancers including PDAC[5], and to be involved in cancer pathogenesis and progression[6]. Reversion of tumor suppressor miRNAs expression to normal levels can restore perturbed cellular homeostasis and activate a therapeutic response[7,8]. Although miRNAs and siRNAs are usually administered separately when tested in cancer animal models and clinical trials, their combination, aiming at various targets, can improve therapeutic efficacy[9].

One of the miRNAs that was associated with good prognosis in PDAC patients[10,11] and also holds a great therapeutic potential[12] is miR-34a. It is a tumor suppressor miRNA downregulated in PDAC[13] which inhibits malignant growth by repressing genes involved in various cellular signaling pathways, such as proliferation, cell cycle, and senescence[14]. Although miR-34a provides prognostic utility, broader molecular signatures that are altered in this cancer might give a better prognosis prediction. To identify additional markers to miR-34a predicting long-term survival with a therapeutic potential, we compared PDAC short-term survivlors (STS <5 months) with long-term survivors (LTS, >2 years) using data from The Cancer Genome Atlas (TCGA). One of the interesting families of cell cycle regulators that exhibited differential expression in LTS versus STS PDAC patients was the serine/threonine Polo-like kinases (PLK), in particular PLK1. The latter, is a mitotic key regulator overexpressed in PDAC patients[15]. Interestingly, a recent study showed that among 38 potential target genes, PLK1 was the only one that distinguished gemcitabine-sensitive versus-resistant pancreatic tumors[16,17].

Following validation of miR-34a and PLK1 reciprocal levels in formalin-fixed-paraffin embedded (FFPE) sections obtained from STS versus LTS PDAC patients, we set to increase miR-34a levels and decrease the expression of PLK1 in a PDAC animal model. We hypothesized that dual delivery of potent synthetic miRNA mimic together with efficacious siRNA might improve therapeutic response. We rationalized to combine miR-34a and PLK1-siRNA in order to attack distinct molecular defects in this cancer while inhibiting MYC, a common target of PLK1[18] and miR-34a[19]. We hypothesized that this approach will lead to a synergistic anticancer effect against PDAC.

Efficient in vivo delivery of miRNA and siRNA for therapeutic purposes is extremely challenging due to low cellular uptake, RNase degradation in the bloodstream, rapid renal clearance, and immunogenicity[20,21]. In order to overcome these limitations of RNAi as anticancer treatment, several non-viral delivery systems have been developed, the majority of them based on a lipidic or polymeric scaffold[21]. Potential novel nanocarriers for the delivery of miRNA/siRNA are poly-(α)glutamic acid (PGA)-based[22,23]. PGA is a promising synthetic polymer with attractive properties: it is water-soluble, non-immunogenic and biodegradable by cathepsin B[24], an enzyme that is highly expressed in most tumor tissues[25]. Furthermore, PGA conjugated to the chemotherapeutic drug paclitaxel (OPAXIO) was shown to be safe at the required doses in clinical trials for the treatment of several cancer types[26–28]. We have recently synthesized a library of aminated polyglutamates for small oligonucleotides complexation[23], out of which a fully aminated polyglutamate backbone was used in vivo for the treatment of ovarian cancer showing promising results[22].

In this study, we further developed a larger globular supramolecular structure based on a PGA backbone for delivering miRNA and siRNA to tumors in vivo. Via the pendent free γ-carboxyl group in each repeating unit of L-glutamic acid of the PGA, we conjugated in parallel ethylenediamine and alkylamine moieties to form a positively charged amphiphilic nanocarrier. Utilizing electrostatic-based interactions, this cationic nanocarrier forms a polyplex with the negatively charged oligonucleotide cargo. The nanocarrier facilitates oligonucleotides delivery by improving their stability in the bloodstream and enabling accumulation of the polyplex at the tumor site due to the enhanced permeability and retention (EPR) effect[29]. Based on our findings from TCGA, we used 2 negatively charged small RNAs: miR-34a for miRNA replacement therapy and PLK1-siRNA for oncogene silencing in an orthotopic PDAC mouse model. We evaluated the formation of therapeutically active nano-scaled polyplexes in pancreatic cancer cells and measured the miRNA mimic-based activity and siRNA silencing achieved in vitro. We further examined the tumor accumulation of the nano-polyplexes carrying miRNA–siRNA combination, their safety profile ex vivo and anticancer efficacy in vivo.

## Results

**High miR-34a/low PLK1 is associated with prolonged survival**. Based on TCGA data obtained from 180 pancreatic cancer patients, whose survival status was available, we found that patients with high miR-34a/low PLK1 expression levels had a significantly longer overall survival (OS) time (longer than 1200 days-LTS) compared to patients with low miR-34a/high PLK1 (<400 days-STS) (log-rank $P < 1.92E-2$) (Fig. 1a). This trend was maintained after controlling for confounders of age, sex, race, and metastasis to lymph nodes (Cox hazard ratio = 2.56, $P < 1.48E-2$). High miR-34a/high PLK1 and low miR-34a/low PLK1 samples were excluded from our survival analysis. To validate these findings, we collected PDAC clinical specimens of STS and LTS (Supplementary Table 1) and evaluated the expression levels of miR-34a and PLK1. Real-time qRT-PCR analysis of miR-34a and immunostaining of PLK1 revealed higher miR-34a levels (Fig. 1b) and lower PLK1 levels (Fig. 1c) in LTS compared to STS. Quantification of PLK1 expression level based on histology scores showed significant difference in PLK1 levels between LTS and STS and also a negative correlation between PLK1 levels and miR-34a levels (Fig. 1d). Since high miR-34a and low PLK1 expression levels correlated with favorable outcomes, we hypothesized that restoration of miR-34a together with silencing of PLK1 could improve the therapeutic response and prolong survival. In order to efficiently deliver this combination of miRNA and siRNA in vivo, we designed and synthesized an amphiphilic polyglutamate amine (APA) polymeric nanocarrier, which is composed of repeating units of PGA.

**APA-miRNA–siRNA nano-polyplexes optimization**. APA nanocarrier was synthesized by subsequently conjugating ethylenediamine and alkylamine moieties to the pending carboxylic groups of the PGA backbone[30]. The resulting polymer ($12{,}513 \text{ gmol}^{-1}$, polydispersity index (PDI) 1.145, Fig. 2a) consisted of 55% positively charged aminated side chains and 45% hydrophobic alkylated side chains. To verify the ability of the polymer to form an electrostatic-based interaction with miRNAs and siRNAs, we incubated several Nitrogen/Phosphate (N/P) ratios of polymer and miRNA–siRNA combination and analyzed the retardation of the small RNAs mobility on agarose gel using

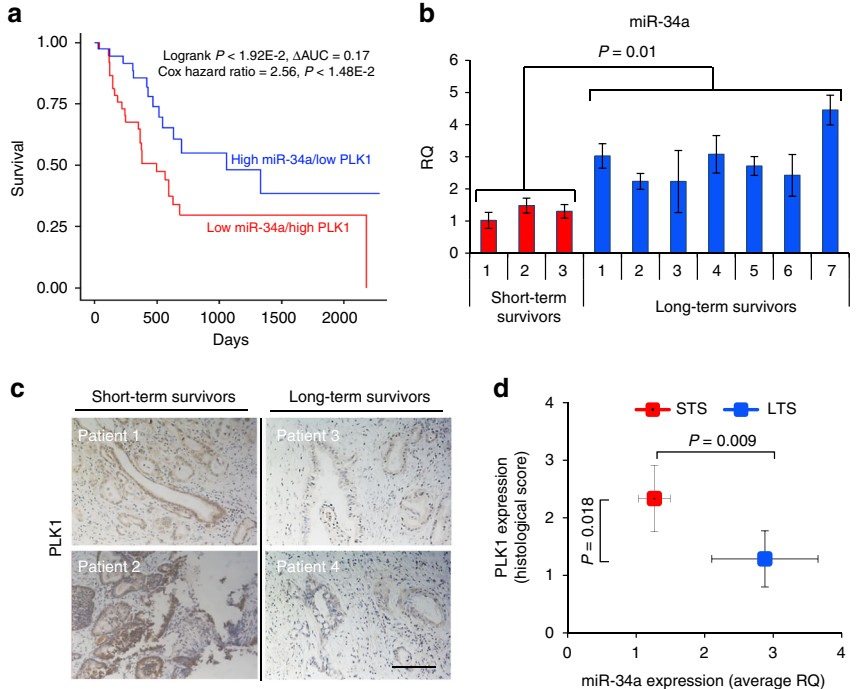

**Fig. 1** High miR-34a/low PLK1 levels are associated with prolonged survival of PDAC patients. **a** Kaplan–Meier curves representing the percent OS in PDAC patients based on combined miR-34a and PLK1 expression levels in the TCGA data set ($n = 180$). Statistical significance between miRNA/mRNA expression and OS was determined by the Log-rank test ($P < 0.05$). **b** High miR-34a levels in FFPE specimens of PDAC patients exhibiting long-term survival (LTS, $n = 7$) compared to patients exhibiting short-term survival (STS, $n = 3$) analyzed by qRT-PCR. **c** Representative images of PLK1 immunostaining from the same PDAC patients showing low PLK1 levels in LTS compared to STS. Scale bar, 100 μm. **d** Quantification of PLK1 immunostaining presented in **c** showing negative correlation to miR-34a levels shown in **b** in PDAC patients. Data represent mean ± SD. (Student's t-test)

electrophoretic mobility shift assay (EMSA). Positively-charged APA was able to form a complex with miRNA–siRNA and neutralize their negative charge with an optimal N/P ratio of 2 (Fig. 2b). The decrease in ethidium bromide fluorescence at high N/P ratio, 4, might indicate strong affinity between the small RNAs and the polymer, resulting in reduced ethidium bromide intercalation. The neutralization of the negatively-charged miRNA–siRNA pair following complexation with the cationic carrier APA was confirmed by surface charge measurements (zeta potential) of the polyplex and found to be almost neutral (4.68 ± 3 mV, Fig. 2c). These polyplexes exhibited spherical structures visible in high-resolution transmission electron microscopy (TEM) (Fig. 2d) with an approximate diameter of 150 nm, which was also confirmed by dynamic light scattering (DLS) measurements (189.79 ± 11 nm, PDI 0.05, Fig. 2c). Following cellular internalization, the small RNA oligonucleotides are expected to be released from the polyplex into the cytoplasm. Therefore, we evaluated the ability of the polyplex to release miR-34a and PLK1-siRNA in the presence of increasing amounts of the polyanion heparin using gel electrophoresis (Fig. 2e and Supplementary Fig. 1). Partial release was obtained at 0.01 heparin Units, while full release was obtained at 1 heparin Unit. Next, we confirmed that the APA-containing polyplex is capable of releasing miRNA following incubation with cathepsin B, a thiol-dependent protease, which degrades PGA and is highly expressed in most tumor tissues[25] including pancreatic cancer[31,32]. Indeed, we observed gradual miRNA release from APA-miR-34a polyplexes over time following incubation with cathepsin B enzyme (Fig. 2f). In order to validate the relevance of polymer degradation by cathepsin B, we profiled the levels of active cathepsins (particularly cathepsin B) in PDAC xenograft tissues. For this purpose, we incubated frozen optimal cutting temperature (OCT) sections of MiaPaCa2 pancreatic tumor

xenograft and tumor-adjacent normal tissue with Cy5-labeled activity-based probe (GB123) that covalently targets active cysteine proteases[33]. As depicted in the fluorescent microscopy images, high-expression levels of active cathepsins were found in MiaPaCa2 pancreatic tumor xenograft tissue, while no considerable expression was observed in the normal adjacent tissue (Fig. 2g). As a control for specificity of labeling, we also incubated MiaPaCa2 tissue with a potent active-site cathepsin inhibitor (GB111)[34] prior to incubation with the Cy5-labeled activity-based probe. No expression of active cathepsins was observed following treatment with this inhibitor (Fig. 2g). We further evaluated whether our cathepsin B-degradable system could be relevant for RNAi delivery in PDAC. Using the Cy5-labeled activity-based probe, we detected active cathepsin B expression in 4 out of 5 primary PDAC cell lines (Supplementary Fig. 2). These results were in accordance with previously described data showing high expression of cathepsins in human PDAC and in human pancreatic intraepithelial neoplasias (PanINs) which are preinvasive precursors of PDAC[35].

**Polyplex internalization into PDAC cells via endocytosis.** We next evaluated the ability of Cy5-labeled siRNA complexed with APA to internalize into human MiaPaCa2 PDAC cells. Confocal images of cellular uptake kinetics of cells incubated with Cy5-siRNA-APA polyplexes for 4, 24, and 48 h showed that the siRNA was taken-up within 4 h with a maximum peak of cellular uptake at 48 h (Fig. 3a, upper panel). Larger magnification of the cells at 48 h following incubation with the polyplexes enabled detection of the predominant accumulation of Cy5-labeled siRNA in the cytoplasm (Fig. 3a, lower panel). In order to further evaluate the cellular localization of the polyplex and to exclude optical artifacts, z-scan analysis was performed (Fig. 3a, lower panel, right).

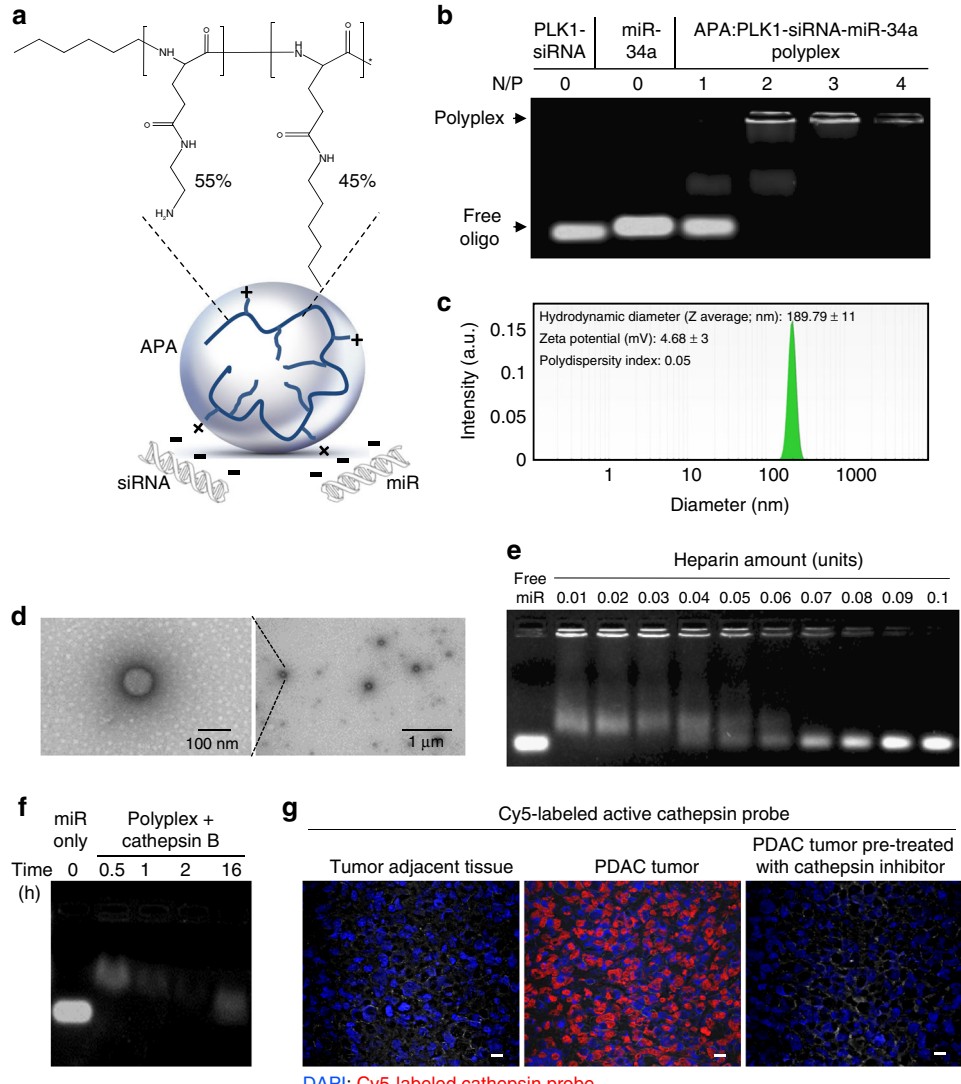

**Fig. 2** Physicochemical characterization of APA-miRNA–siRNA polyplexes. **a** Schematic illustration of APA nanocarrier complexed with small RNAs (polyplex) and chemical structure of APA polymeric nanocarrier. **b** Polyplex formation of APA with miR-34a and PLK1-siRNA (total of 50 pmol oligonucleotides, miRNA/siRNA ratio of 1:1) showing the optimal Nitrogen/Phosphate (N/P) ratio using EMSA. **c** Hydrodynamic diameter and surface charge of the polyplex at N/P ratio of 2, measured by particle size analyzer and Zetasizer, respectively. **d** Representative TEM images of the polyplex. **e** miR-34a release from the polyplex obtained in vitro by the polyanion heparin displacement assay. **f** miR-34a release from the polyplex by cathepsin B (2 units per mg polymer) cleavage of the PGA backbone. **g** Direct labeling of active cathepsins in PDAC tumor xenograft and in normal adjacent tissues. Frozen sections were fixed on slides, incubated with 0.25 μM Cy5-labeled cathepsin activity-based probe (in red), stained with 4′,6-diamidino-2-phenylindole (DAPI, in blue) and imaged with fluorescent microscope. For specificity of staining, additional slides were treated with a non-labeled cathepsin inhibitor (GB111, 5 μM) prior to incubation with the Cy5-labeled cathepsin activity-based probe (right image). Scale bar, 10 μm

The siRNA was detected at the same focal plane as the nuclei, confirming its intracellular localization. Next, cellular internalization of APA-siRNA polyplex was evaluated in live cells using imaging flow cytometry which revealed that APA was capable of delivering Cy5-siRNA into $42.47 \pm 1.4\%$ of the cancer cells (Cy5-positive), as compared to cells treated with Cy5-siRNA alone with only 0.03% Cy5-positive cells (Fig. 3b). Transfection using Lipofectamine 2000 served as positive control for siRNA internalization with $32.65 \pm 1.3\%$ Cy5-positive cells. The amount of cells that internalized the polyplex (shown also by the internalization histograms, Fig. 3b, lower panel) prompted us to further investigate the intracellular uptake, revealing two distinct cell morphologies of MiaPaCa2 displaying a different pattern of siRNA uptake. According to the ATCC, MiaPaCa2 cells exhibit two alternative morphologies: one is adherent epithelial cells and the second is floating round cells. As shown in Supplementary

Fig. 3, Cy5-labeled siRNA-APA polyplexes were successfully internalized into the adherent, larger cell population in size (R4), and not to the round, smaller in size cell populations (R3 and R2). We observed the same intracellular uptake pattern using Lipofectamine 2000 as transfection reagent. Thus, we concluded that in live MiaPaCa2 cells siRNA uptake is achieved mostly by the adherent large cell population. In an attempt to understand whether this phenomenon is unique to MiaPaCa2 cells, we sought to evaluate the transfection efficiency in additional pancreatic cancer cell lines. Flow cytometry analysis showed that Cy5-labeled APA complexed with siRNA successfully transfected $86.64 \pm 4.4\%$ of cells derived from the transgenic Kras$^{G12D}$; Trp53$^{R172H}$; Pdx1-Cre mouse (KPC), $97.43\% \pm 3.07$ of Panc02, $46.76\% \pm 2.7$ of Panc1 and $93.78\% \pm 4.6$ of BxPC3 cells (Supplementary Fig. 4). Next, we aimed to gain insight into the intracellular trafficking of APA-siRNA polyplex. Confocal

microscopy analysis followed by immunostaining with early endosome antigen 1 (EEA1) and with lysosome-associated membrane protein 1 (LAMP1), showed that after 48 h incubation, the majority of the internalized polyplexes were not co-localize with early endosomes or with late endosomal/lysosomal compartments (Fig. 3c). Furthermore, the percentage of poly-plexes that did not co-localized with endosomal/lysosomal compartments gradually increased over time (from 47 to 73%,

Fig. 3d). In contrast, the percentage of polyplexes which co-localized with the early endosome decreased over the same period of time (from 36 to 17%, Fig. 3d). This might be explained by a time-dependent release of polyplexes from early endosomes into the cytoplasm. Co-localization of polyplexes with lysosomes was relatively low (~10%) and hardly changed during this time course. Polyplex internalization via endocytosis 4 h following incubation is depicted in Fig. 3e and Supplementary Movie 1.

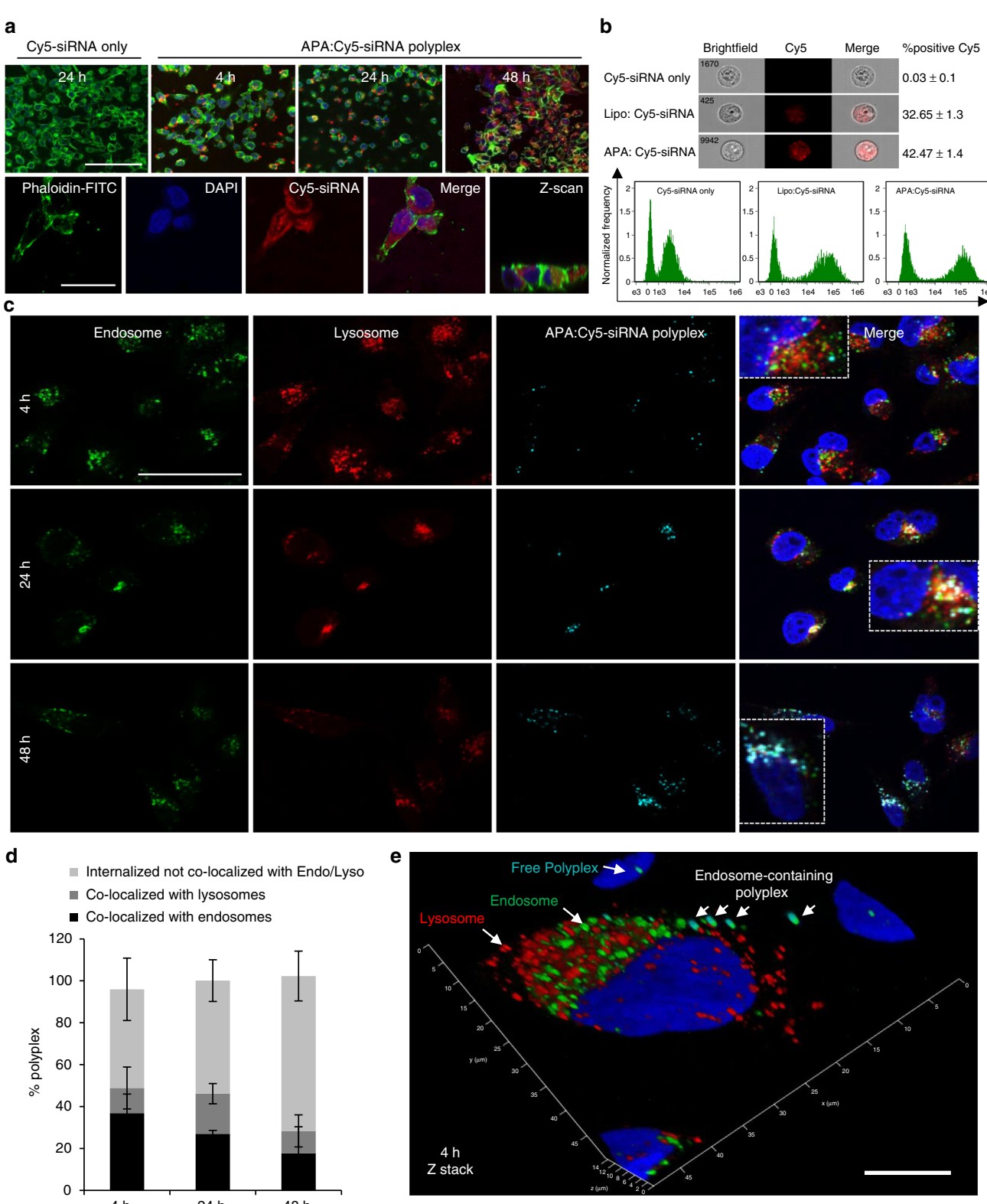

**In vitro synergistic antitumor effect of RNAi combination**. To confirm in vitro miRNA and siRNA delivery efficacy, APA polymer carrying separately, miR-34a or PLK1-siRNA, at the optimal N/P ratio of 2, was applied to cultured MiaPaCa2 cells and the levels of miR-34a and its target genes, as well as the levels of PLK1 mRNA and protein were quantified using real-time qRT-PCR and western blot analyses. A significant increase in miR-34a levels following transfection with the APA-miR-34a polyplexes was observed compared to the untreated cells and NC-miR-treated cells, with *circa* 900-fold change increase after 72 h (Fig. 4a). miR-34a delivered by APA was active and potently downregulated its target genes Notch1, CDK6, Bcl2, and MET protein levels (Fig. 4b). Downregulation of these target proteins ranged between 44% and 84%. Transfection with a non-targeted negative control (NC)-miR had no effect on the expression levels of the evaluated target genes. Transfection of cells with APA-PLK1-siRNA polyplexes was also efficient and decreased PLK1 expression by 50 and 80% at the mRNA and protein levels, respectively (Fig. 4c, d).

miR-34a was identified as having a tumor suppressive function in pancreatic cancer, in which it is commonly deleted[36]. siRNA targeting PLK1 was found to inhibit the proliferation of pancreatic cancer cells[37]. Therefore, we determined the ability of those two anticancer candidates, miR-34a and PLK1-siRNA complexed with our APA nanocarrier, to inhibit the tumorigenicity of pancreatic cancer cells. To that end, we measured cell viability (Fig. 4e–g), cell migration (Fig. 4h), and cell growth and survival using a clonogenicity assay (Fig. 4j) following treatment with APA-miRNA–siRNA polyplexes. First, cells were transfected with serial concentrations of APA-polyplexes containing miR-34a or PLK1-siRNA alone and viable cells were counted by Coulter Counter. Both miR-34a alone (Fig. 4e) and PLK1-siRNA alone (Fig. 4f) significantly reduced the viability of MiaPaCa2 cells in a dose-dependent manner (up to 49% at 250 nM miRNA concentration and up to 58% at 100 nM siRNA concentration) compared with the NC-miR/NC-siRNA treatment and with the untreated cells. Moreover, the combination treatment of miR-34a and PLK1-siRNA further reduced cell viability compared to each RNAi alone (Fig. 4g). To evaluate the nature of this reduction in cell viability, we used the additive model[38]. Using this method, we determined a synergistic reduction in the combined treatment (observed/expected ratio = 0.52, Supplementary Table 2).

Cell migration was evaluated in a wound healing assay using the IncuCyte Live Cell Analysis System. Cells treated with NC-miR-NC-siRNA and cells that were left untreated closed the wound almost completely (~80%) within 48 h (Fig. 4h, Supplementary Fig. 5). Cells treated with miR-34a or PLK1-siRNA partially closed the wound over this time frame, whereas cells treated with the RNAi combination closed only 50% of the wound (a reduction of 30% compared to the untreated cells and to the NC-miR-NC-siRNA treated cells) (Fig. 4h, i, Supplementary Fig. 5 and Supplementary Movie 2). The effect on growth and

survival of pancreatic cancer cells was assessed also using clonogenic assay. MiaPaCa2 cells that were treated with the combination of miR-34a and PLK1-siRNA using APA had low number and small size of surviving colonies (by 64%) compared to the untreated cells (Fig. 4j, k).

In order to validate that our therapy is applicable not only for MiaPaCa2 cells, we set to evaluate its ability to inhibit cell proliferation and colony formation on two additional human PDAC cell lines, BxPC3 and Panc1 and on the murine cell line, KPC. The latter is derived from the genetically engineered mouse model (GEMM) KPC, which was shown to be of relevance to the human clinical set-up[39]. KPC mice conditionally express endogenous mutant Kras and p53 alleles in pancreatic cells[40,41], which develop pancreatic tumors whose pathophysiological and molecular features resemble those of human PDAC. The combination treatment showed synergistic reduction in cell viability (of 72% in KPC, 70% in Panc1 and 65% in BxPC3) as analyzed using the additive model (Supplementary Fig. 6a, b, Supplementary Fig. 7a–d). In addition, KPC cells treated with the combination closed only 33% of the wound compared to the untreated cells and to the NC-miR/NC-siRNA treated cells that closed the wound almost completely (Supplementary Fig. 6c). The combination significantly reduced the number and size of surviving colonies of PDAC cells (by 72% in KPC, 75% in Panc1, and 55% in BxPC3) compared to the untreated cells (Supplementary Figs. 6d, 7e, f). These results indicate that APA carrying miR-34a/PLK1-siRNA combination could inhibit pancreatic cancer cells growth, migration, and survival effectively in a synergistic manner.

**APA-siRNA polyplexes demonstrate a biocompatible profile**. In order to evaluate the safety profile of APA as a nanocarrier, an ex vivo cytokine-induction assay was performed using human peripheral blood mononuclear cells (PBMCs), which determined the secretion level of inflammatory cytokines, such as IL-6 and TNF-α representing the innate immune response. Neither APA alone nor APA-siRNA polyplex promoted the secretion of the evaluated cytokines (Fig. 5a) compared to the Toll-like receptor 4 natural ligand, lipopolysaccharides (LPS), that induced the secretion of high levels of both cytokines. Next, in order to determine the ability of the polyplex to be administered systemically, stability of the polyplex was evaluated in vitro in 100% fetal bovine serum (FBS). There was no release of miRNA from the polyplexes prepared at N/P ratio of 2 up to 12 h of incubation, while polyplexes of APA and siRNA prepared at 1.5 N/P ratio demonstrated much lower stability[30]. Moreover, APA polymer stabilized miR-34a and prevented its degradation for a longer time (12 h) compared to naked miR-34a (Fig. 5b). Biocompatibility was also assessed by measuring red blood cell (RBC) lysis. Serial dilution of APA (1−10,000 µg ml$^{-1}$) were used at the relevant in vivo concentrations and were adjusted to their dilution in

**Fig. 3** Cellular internalization and trafficking of APA-siRNA nano-polyplexes in MiaPaCa2 cells. **a** Confocal images of MiaPaCa2 cells incubated with APA: Cy5-labeled siRNA polyplexes (in red) for 4, 24, and 48 h showing a maximum peak of cellular uptake at 48 h (upper panel). Cy5-labeled siRNA alone was used as control (upper panel, left). Actin filaments are in green and nuclei are in blue. Lower panel is a larger magnification of a representative field following 48 h incubation with the polyplex showing predominant accumulation of siRNA in the cytoplasm. **b** Brightfield and fluorescent images of live MiaPaCa2 cells showing Cy5-labeled siRNA delivery by APA nanocarrier compared to delivery by Lipofectamine 2000, analyzed by Imaging Flow Cytometry. Cy5-labeled siRNA alone served as control. Lower panel shows the Cy5 internalization histograms. **c** Intracellular trafficking of APA:Cy5-labeled siRNA polyplexes (100 nM siRNA, light blue) in MiaPaCa2 cells at different time points showing internalization via endocytotic pathway. Cells were stained with early endosome marker EEA1 (green) or late endosome/lysosome marker LAMP1 (red). Nuclei were stained with DAPI (blue). **d** Quantitative analysis of **c** showing co-localization of APA:Cy5-labeled siRNA polyplexes with EEA1 and LAMP1 4, 24, and 48 h following polyplex incubation. Data represent mean ± SD of 7 random fields. **e** A z-stack confocal image at 4 h after incubation with the polyplex showing endosomes, lysosomes, endosome-containing polyplexes, and polyplexes inside a single cell. Endo; Endosome, Lyso; Lysosome. Scale bar; **a**- upper panel, 100 µm; **b**- lower panel, 25 µm; **c**- 50 µm; **e**- 10 µm. All confocal experiments were done 2–3 times and data are presented as mean ± SD

the mouse blood volume (2 ml). The results clearly showed that at these concentrations our polyplexes did not cause hemolysis ex vivo and are, therefore, suitable for intravenous (i.v.) administration (Fig. 5c). To determine the maximum tolerated dose (MTD) of APA-miRNA–siRNA polyplexes, we evaluated the viability of Balb/c mice and monitored them for a period of

5 weeks, following a single i.v. injection of the polyplex at various small RNA oligonucleotide doses of miRNA–siRNA (1, 2, 4, and 6 mg kg$^{-1}$, at 1:1 ratio). The mice were viable following all miRNA–siRNA doses (Table 1). We further assessed the polyplexes effect on mouse normal pancreas and on glucose levels in mouse blood. Following 3 sequential i.v. injections of APA-miR-

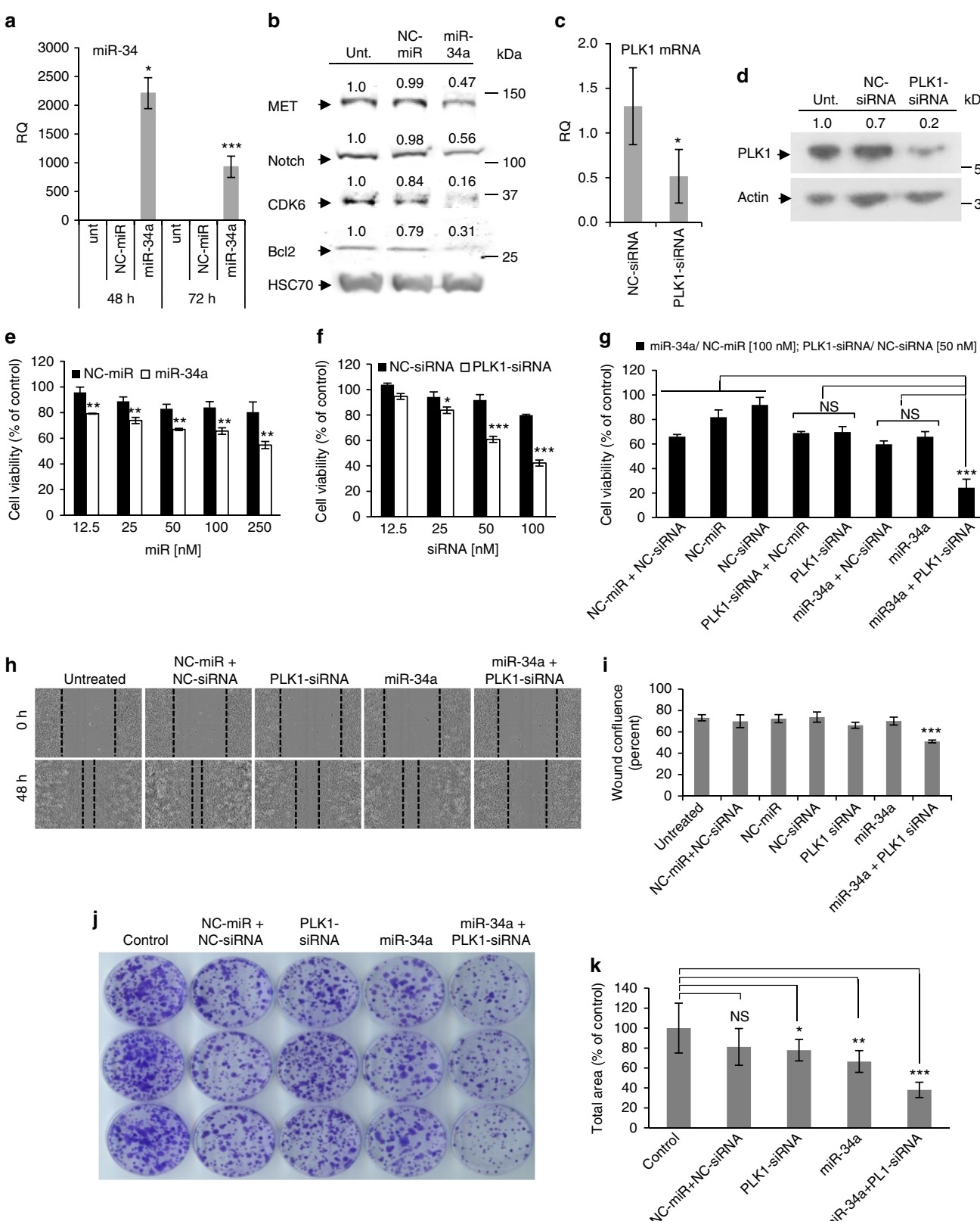

34a-PLK1-siRNA polyplexes or PBS, blood glucose levels were measured from the tail vein and pancreas was resected, embedded in paraffin and stained with Hematoxylin and Eosin (H&E). No differences in normal pancreas morphology as well as in blood glucose levels were observed between PBS-treated and polyplex treated mice (Fig. 5d, e). Taken all together, we demonstrated the safety profile of our nanocarrier.

**Polyplexes selectively accumulate in orthotopic PDAC tumors.** To assess whether APA nanocarrier accumulates preferably at the tumor site once injected systemically, we developed an orthotopic pancreatic cancer mouse model by injecting mCherry-labeled MiaPaCa2 cells into the pancreas of SCID mice. Two-weeks post injection, tumor growth was monitored by the increase in mCherry fluorescent signal (Supplementary Fig. 8a, b) which was found only in the pancreas (Supplementary Fig. 8c). In addition, we characterized the pancreatic tumor vasculature functionality and morphology of our orthotopic xenograft mouse model, which showed enlarged unorganized leaky blood vessels as compared to normal pancreas (Supplementary Fig. 10a–d). To determine whether our orthotopic PDAC mouse model mimics properly the clinical setting of dense stroma, FFPE sections were stained for α-Smooth Muscle Actin (α-SMA) for activated fibroblasts. Indeed, cancer-associated activated fibroblasts were highly visible in PDAC compared to healthy pancreas (Supplementary Fig. 10e). To study the pharmacokinetics profile of polyplexes, mice bearing orthotopic mCherry-labeled tumors were administered via the tail vein with APA: Cy5-labeled siRNA polyplexes (0.5 mg kg$^{-1}$ siRNA, 100 μl) or with Cy5-labeled siRNA alone at the same dose and imaged at different time points. The polyplex demonstrated preferable accumulation in the tumor site over time up to 24 h as shown by the Cy5 fluorescent signal in the mice treated with APA:Cy5-siRNA (Fig. 6a). When Cy5-siRNA was injected without APA, no accumulation was observed in the intact mice (Fig. 6a). In order to determine the biodistribution profile of the polyplex, we resected the tumors and healthy organs (heart, lungs, liver, kidneys, and spleen) from mice, 24 h following i.v. administration of Cy5-labeled siRNA alone or complexed with APA and measured the fluorescent intensity of Cy5. When Cy5-labeled siRNA was injected while complexed with APA nanocarrier, there was accumulation of the fluorescent signal in the tumor and relatively low accumulation in the kidneys, spleen, heart, lungs, and liver (Fig. 6b, Supplementary Fig. 9). In contrast, when Cy5-labeled siRNA was injected alone, we detected accumulation mainly in the kidneys and relatively small amounts in the tumor (Fig. 6b). Quantification of Cy5 component revealed a significant fivefold increase in total signal (scaled counts s$^{-1}$ g$^{-1}$ tissue) of the siRNA complexed with APA compared to siRNA alone at the tumor site (Fig. 6c). There was no significant difference in the heart, lungs, liver, and kidneys localization between treatments

with free or complexed RNAi. Further confocal analysis of frozen samples prepared from the resected tumors confirmed that the polyplex accumulated in the PDAC tumor (Fig. 6d). We further assessed the accumulation of miR-34a in the orthotopic PDAC tumors following 3 sequential i.v. injections of PBS or polyplex formulated with miR-34a or NC-miR ($n = 4$ mice per group, 2 mg kg$^{-1}$ miR dose). miR-34a levels were fivefold higher in tumors isolated from mice treated with APA-miR-34a polyplex, compared to mice treated with either PBS or APA-NC-miR (Fig. 6e). We also quantified miR-34a target genes levels by real-time qRT-PCR. Tumors from APA-miR-34a-treated mice had reduced levels of Bcl2, CDK6, MET and Notch1 by 45, 25, 20, and 11%, respectively, relative to APA-NC-miR treated mice (Fig. 6f). These data suggest that APA nanocarrier successfully delivered functionally therapeutic small RNA oligonucleotides into PDAC tumors.

**In vivo antitumor effect of miRNA–siRNA combination.** To determine the therapeutic effects of our combined miR-34a and PLK1-siRNA treatment in vivo, tumor-bearing mice were randomized into four nanoparticle treatment groups: (i) miR-34a/PLK1-siRNA, (ii) miR-34a/NC-siRNA, (iii) PLK1-siRNA/NC-miR, and (iv) NC-miR/NC-siRNA, and a group treated with PBS ($n = 6/7$ mice per group). Mice were i.v. injected as depicted in Fig. 7a. Tumor growth monitoring using intravital fluorescence imaging revealed that miR-34a/PLK1-siRNA combination therapy induced pancreatic tumor regression, inhibiting tumor growth to an average of 3.85% (111.3 ± 39 scaled counts per s) compared to tumors treated with PBS or NC-miR/NC-si ($P < 0.05$ for both) at day 45 (Fig. 7b). Each monotherapy of miR-34a/NC-siRNA or PLK1-siRNA/NC-miR inhibited tumor growth to an average of 31% (896.6 ± 91 scaled counts per second) and 44.25% (1278.0 ± 459 scaled counts per second), respectively (Fig. 7b). All animals tolerated small RNA therapy well, with no significant weight loss (Fig. 7c). A representative image of treated mice at day 33 showed differences in fluorescence signal (Fig. 7d). This suggests that targeted combination RNAi therapy using miR-34a and PLK1-siRNA elicit a potent antitumor response. Survival of mice treated with the combination was significantly prolonged ($P < 0.05$) compared to all other treatment groups (Fig. 7e). We then evaluated the effect of our combination therapy on proliferation, apoptosis, and angiogenesis on FFPE sections of tumors resected from treated mice on day 45. For that, we immunostained the resected tumors with ki67, cleaved caspase 3 and CD31 antibodies. We observed that the combination treatment inhibited proliferation and angiogenesis and increased apoptosis to a larger extent compared to the monotherapies (Fig. 7f, g). The effects of the treatments on blood counts and chemistry were also studied. For that, blood was withdrawn from mice treated with either PBS or APA-small RNA oligonucleotides

**Fig. 4** Proliferation, migration and survival inhibition of MiaPaCa2 cells following treatment with APA nanocarrier containing miRNA–siRNA combination. **a** miR-34a levels in MiaPaCa2 cells following treatment with APA polyplexes containing miR-34a or NC-miR, quantified relative to U6 RNA using qRT-PCR, showing in vitro delivery efficacy by APA nanocarrier. Untreated cells served as control. **b** miR-34a's target genes (CDK6, MET, Notch, and Bcl2) levels 48 h following the same treatment as in **a**. **c** PLK1 mRNA levels following treatment with APA polyplexes containing PLK1-siRNA or NC-siRNA for 24 h, quantified relative to GAPDH mRNA using qRT-PCR. **d** PLK1 protein levels following the same treatment as in **c**. Densitometric analysis of western blot is presented as percentage of band intensity compared to untreated cells (unit). **e–g** Proliferation of MiaPaCa2 cells following treatment with APA polyplexes containing different concentrations of miR-34a or NC-miR **e**, PLK1-siRNA or NC-siRNA **f**, or miR-34a (100 nM) and PLK1-siRNA (50 nM) combination **g**. Statistical significance is shown for the comparison between miR-34a and NC-miR in **e** and between PLK1-siRNA and NC-siRNA in **f** ($n = 3$). **h, i** Migration of MiaPaCa2 cells 48 h following incubation with the same treatments as in **g**. Representative images of the cells from 0 to 48 h time points **h** quantified as wound confluence (percent out of initial wound at time 0) using the IncuCyte software **i**. (2 biological repeats were done in triplicates). **j** Representative images of cell survival via colony formation assay for 11 days. **k** Quantification of colonies from 3 biological repeats as their total area relative to untreated cells (control) using ImageJ software (experiments were done in triplicates). Data represent mean ± SD. (Student's t-test, NS; not significant for $P > 0.05$, *$P < 0.05$, **$P < 0.01$, ***$P < 0.001$.)

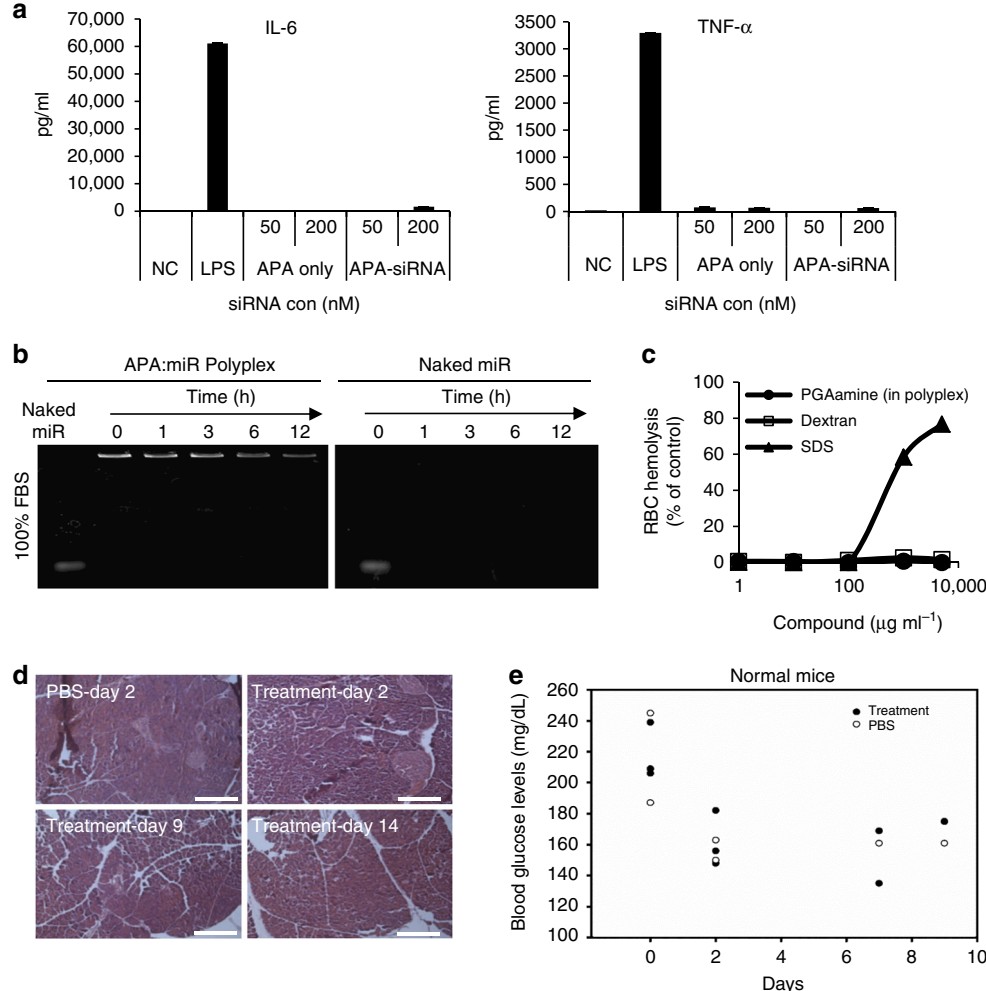

**Fig. 5** Biocompatibility of APA-siRNA polyplex. **a** APA alone or complexed with siRNA (50 and 200 nM) was added to freshly isolated human PBMCs that were seeded in 12-well plates. PBMCs medium and LPS (2 µg ml$^{-1}$) were used as negative and positive controls, respectively. Culture supernatants were collected after 24 h and assayed for human IL-6 and TNF-α cytokines by ELISA. **b** miR (35 µM) alone or complexed with APA was incubated in FBS for several time points (0, 1, 3, 6 and 12 h) at 37 °C and was run on an electrophoresis agarose gel. **c** Red blood cells lysis assay following 1 h incubation with APA-miRNA polyplexes. Results are presented as percentage of hemoglobin released by 1 wt %/vol solution of Triton X-100 (100% lysis). Sodium dodecyl sulfate (SDS) and dextran were used as positive and negative controls, respectively. **d** H&E staining of normal pancreas following 3 sequential i.v. injections of PBS or APA-miR-34a-PLK1-siRNA polyplex (2 mg kg$^{-1}$ oligonucleotide dose). Scale bar, 10 µm. **e** Blood glucose levels of normal mice treated as in **d**. Blood was withdrawn at days 0, 2, 7, and 9 from treatment initiation (n = 2/3). Data represent mean ± SD

| Table 1 Maximum tolerated dose of APA:siRNA-miRNA polyplex | | | | | |
|---|---|---|---|---|---|
| N/P ratio | siRNA dose (mg kg$^{-1}$) | miRNA dose (mg kg$^{-1}$) | Total RNA dose (mg kg$^{-1}$) | Polymer dose (mg kg$^{-1}$) | Survival |
| 2 | 3 | 3 | 6 | 16.07 | + |
| | 2 | 2 | 4 | 10.7 | + |
| | 1 | 1 | 2 | 5.3 | + |
| | 0.5 | 0.5 | 1 | 2.6 | + |

on day 40 from tumor inoculation. None of the treatments affected blood counts (Supplementary Fig. 11) nor blood chemistry parameters (Supplementary Table 3).

**Antitumor effect is due to a decrease in the shared target MYC.** We hypothesized that the superior antitumor response following combination treatment might be explained by targeting important oncogenes downstream to miR-34a and PLK1. To identify these shared targets, we looked for transcripts that are predicted as targets of both of them. Using miRNA target prediction and

molecular pathways analyses, we identified V-Myc Avian Mye-locytomatosis Viral Oncogene Homolog (MYC) as a miR-34a target (miRanda; mirSVR score: −0.163) (Fig. 8a), as well as a downstream protein to PLK1. Previous work has already confirmed using a reporter assay that miR-34a directly targets MYC through a conserved seed region in its 3′-UTR at positions 123–144[19]. Consistent with this, following ectopic expression of miR-34a alone or combined with PLK1 using our polymeric nanocarrier, the expression of MYC was reduced in MiaPaCa2 cells (Fig. 8b). Furthermore, immunohistochemistry staining of tumor tissues from the in vivo efficacy study showed that the

combination therapy significantly reduced MYC levels (Fig. 8c). To understand the clinical relevance of the observed down-regulation of MYC, we evaluated the levels of MYC in the same FFPE specimens of short-term and long-term PDAC patients presented in Fig. 1. MYC immunostaining of these samples revealed low levels of MYC in LTS compared to STS (Fig. 8d, e). We also evaluated the levels of other target genes of miR-34a,

MET and Bcl2, in these patient samples. We found no significant difference in MET and Bcl2 expression levels between the STS and the LTS (Supplementary Fig. 12). To further verify our hypothesis that MYC is the mediator of the antitumor properties of our therapy, we evaluated whether overexpression of MYC could rescue the tumorigenic phenotype of PDAC cells. Mia-PaCa2 cells were transiently transfected with MYC-ORF plasmid

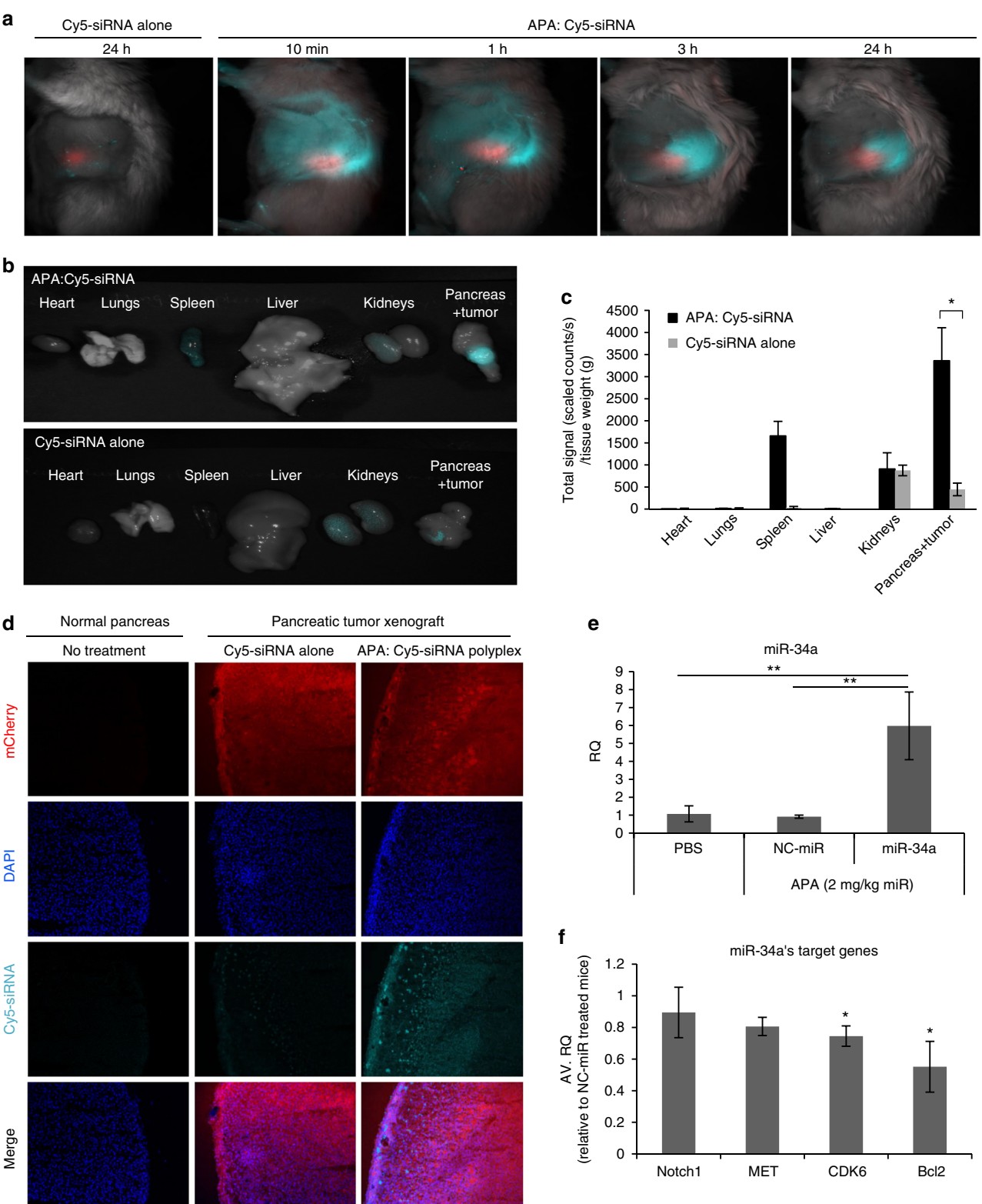

that lacks the 3′-UTR and thus miR-34a binding site. Following MYC overexpression which was confirmed by immunoblotting (Fig. 8f), cells were treated with miR-34a-PLK1-siRNA combination or left untreated. Cell viability of naive (wild type) Mia-PaCa2 PDAC cells was decreased when treated with the combination. However, MYC-overexpressed cells treated with the combination showed no significant difference in cell viability compared to MYC-overexpressed-cells that were not treated (Fig. 8f). These results suggest that the effect of the combination of miR-34a and PLK-siRNA is dependent on MYC down-regulation. According to these results, we suggested a model for the mechanism showing how our dual miRNA–siRNA combination treatment contributes to the synergistic anticancer effects in PDAC via MYC (Fig. 8g).

## Discussion

In this study, we initially identified a combination of prognostic markers, miR-34a and PLK1, that showed better OS in pancreatic cancer patients. We hypothesized that by using this specific combination, we could attack distinct and central molecular pathways in pancreatic cancer, which in turn will lead to improved therapeutic response and prolonged survival. We show here the rational synthesis and characterization of a novel bio-degradable PGA-based polymeric nanocarrier for the combined delivery of siRNA and miRNA to PDAC tumors. Systemic in vivo delivery of miR-34a and PLK1-siRNA by our APA nanocarrier, which facilitated accumulation preferably at the tumor site and internalization into the tumor cells, efficiently inhibited tumor growth in an orthotopic mouse model, with no systemic side effects nor immunotoxicity. This delivery of the combined therapy synergistically improved preclinical outcomes, leading to significant reduction in primary tumor growth and hence might be considered in the future as a possible addition to conventional chemotherapeutic drugs given in the clinic. One major advantage of our delivery system is its ability to carry any combination composed of RNA or DNA oligonucleotides needed for the same cell for optimal synergistic efficacy. This type of nanocarrier-cargo system should enable loading of specific therapeutic agents for tailor-made treatments.

Since small RNAs can stimulate innate cytokine responses, which may be amplified by efficient delivery into the cell by carriers[42], it is crucial to assess the carrier's immunogenic potential. Whereas our APA nanocarrier did not induce IL-6 and TNF-α cytokine secretion, complement activation of the immune response will require further evaluation. Another important pre-requisite for successful delivery of oligonucleotides in vivo is the stability of the polyplex under the biological milieu. We showed that our polyplex was stable in serum following 12 h of incubation and improved the stability profile of miRNA compared to its naked form. This could be attributed to steric hindrance created by the polymer and to the stable complex formed between the RNA and the polymer, thereby decreasing its interaction with exogenous RNases present in the serum. Future studies will need to explore the use of a further stabilized polyplex to increase the

circulation time in the bloodstream beyond 12 h in order to extend the long-term effect of the treatment.

Polymer-RNAi polyplexes accumulate selectively at tumor tissues by the EPR effect. The impaired hyperpermeable tumor vessels, as was demonstrated in our orthotopic mouse model (Supplementary Fig. 10), allow preferential extravasation of circulating macromolecules that are subsequently retained there due to poor lymphatic drainage[43]. Use of nano-sized polymeric carriers enables passive targeting of miRNAs and siRNAs to tumors, minimizing non-specific targeting to healthy organs and lowering the amount of small RNAs that needs to be administered to reach the desired therapeutic effect. We showed that APA-siRNA polyplexes accumulated selectively at the tumor site, following systemic administration. However, a small amount of polyplexes accumulated also in the spleen. This could be explained by the fact that upon systemic administration, nano-sized carriers are rapidly distributed to organs in the reticuloendothelial system (RES) and phagocytosed by the mononuclear phagocyte system (e.g., macrophages and liver Kupffer cells). These clearance processes, mediated by the interaction of particles with blood components (e.g., immunoglobulins of the complement system), result in higher particle accumulations in RES organs, such as liver and spleen[44]. Both polyplex and siRNA alone were accumulated, to the same extent, in the kidneys. This is in accordance with previously shown accumulation of radiolabeled siRNA in the kidneys of rats which was excreted via the urine, post i.v. injection[45].

Although each of the monotherapies, miR-34a and PLK1-siRNA, was tested alone in the clinic, their combination has not been evaluated yet. Indeed, human PLK1 was found to be involved in the formation and progression of many tumor types[46]. To date, PLK1-siRNA is one out of four RNAi-based drugs that have been evaluated in early clinical trials for cancer therapy through systemic administration[47]. miR-34a has also entered clinical trial lately in order to test its safety in primary liver cancer and other selected solid tumors or hematologic malignancies (ClinicalTrials.gov identifier: NCT01829971). Here, we evaluated, for the first time, the ability of the combination of these two anticancer RNAi candidates complexed with our APA nanocarrier to inhibit pancreatic cancer progression. We found significant reduction in cell viability, migration and survival following treatment with APA-miRNA–siRNA combined polyplexes. Although we showed in vivo efficacy only in the human MiaPaCa2 model, we found convincing evidence from the literature showing both overexpression of PLK1 oncogene and downregulation of miR-34a in PDAC patients. Pancreatic adenocarcinomas were PLK1 positive in 47.7% of cases (out of 86 patient samples) in one study[15] and in 80% of cases (out of 140 patient samples) in another[48]. Also, the expression of miR-34a was shown to be frequently lost in 15 pancreatic cancer cell lines compared to normal pancreatic cell lines (HPNE and HPDE)[14]. Taken together, we presented the potential of our combination therapy as clinically relevant for PDAC patients.

Based on TCGA data and our results from several human and murine cell lines, we can determine that our miR-34a-PLK1-

**Fig. 6** Biodistribution and accumulation of APA-miRNA–siRNA polyplexes in orthotopic pancreatic tumor-bearing mice. **a** APA:Cy5-labeled siRNA polyplexes or Cy5-labeled siRNA alone (0.5 mg kg$^{-1}$ siRNA dose) were injected i.v. to mCherry-labeled tumor-bearing (~1000 scaled counts per second) mice. At 10 min, 1, 3, and 24 h mice were imaged by non-invasive intravital fluorescent microscopy for mCherry (red) and Cy5 (light blue) fluorescent signals. Representative images of the mice are shown (n = 3). **b**, **c** Twenty-four hours following intravenous injection of the same treatments as in **a**, tumor and healthy organs were resected, imaged **b** and quantified for their Cy5 fluorescent signal intensity **c** (n = 3). **d** Resected tumors were embedded within OCT, cut to 10 μm sections, stained with DAPI and subjected to confocal microscopy. Normal pancreas was used as control. Scale bar, 250 μm. **e** Relative miR-34a levels in PDAC tumors following intravenous injections (3 consecutive, once a day) of APA-miR-34a or APA-NC-miR (2 mg kg$^{-1}$ miR dose) or PBS, quantified by qRT-PCR (n = 4). **f** miR-34a target genes level following injection of the same treatments as in **e**, quantified by qRT-PCR (n = 4). Data represent mean ± SEM in **c** and mean ± SD in **e** and **f**. (Student's t-test, *P < 0.05, **P < 0.01)

siRNA APA treatment is clearly relevant to patients with inverse correlation expression of low miR-34a and high PLK1. However, as high and low levels are relative to an unknown control in the clinic, it seems to us that in the case of PDAC, upregulating miR-34a and downregulating PLK1 will probably be beneficial even if

both markers are low or both are high. As noted in the beginning, we compared survival between high miR-34a/low PLK1 and low miR-34a/high PLK1 in order to identify the feasibility of our intervention to affect patient survival. In fact, we observe that miR-34a and PLK1 expression is marginally positively correlated

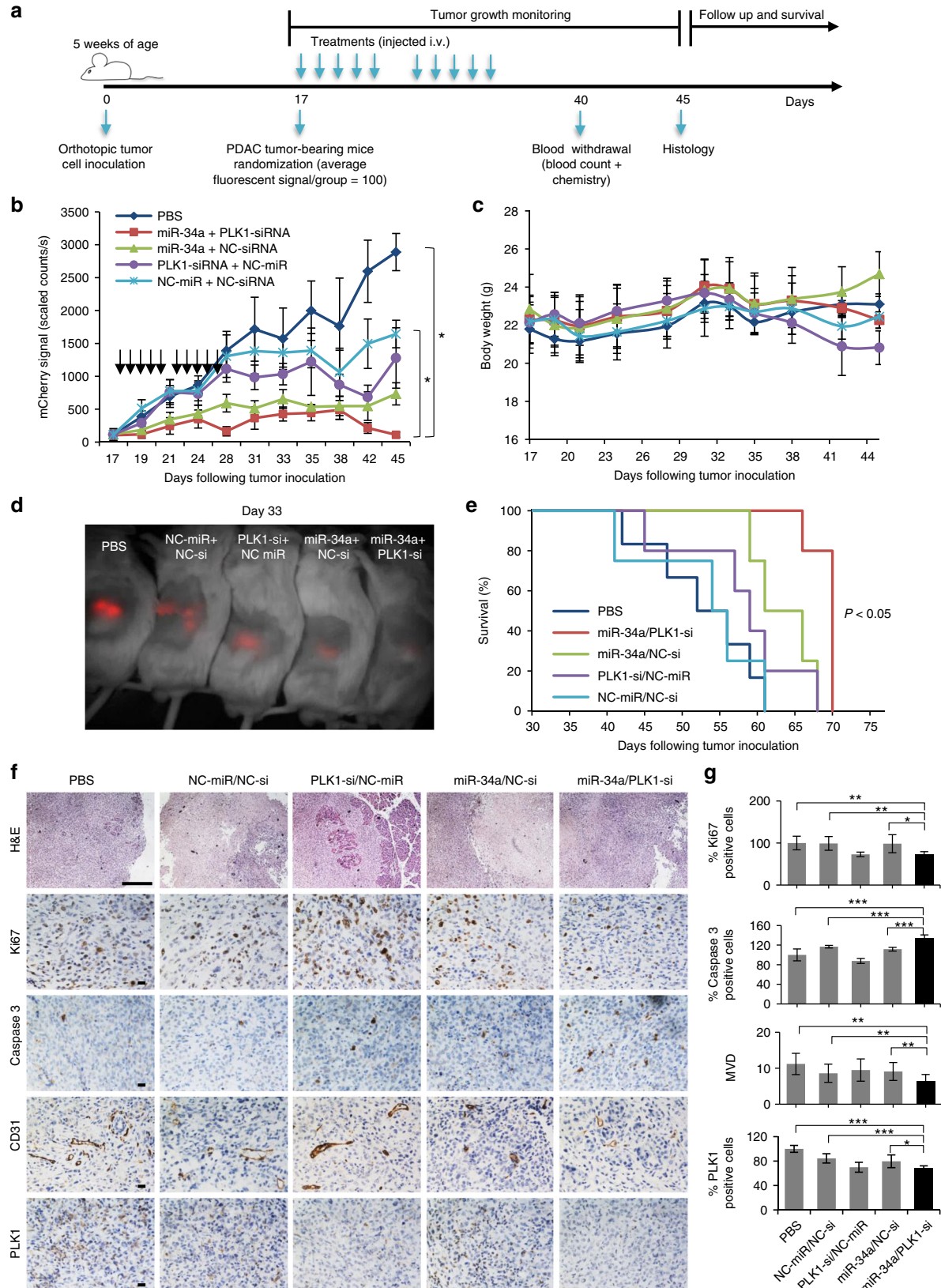

(Spearman $R = 0.19$, $P < 0.01$), thus our intervention is likely to revert this correlation, lower the tumor's fitness, and lead to improved patient survival. To prove this, we will need to evaluate our system on an array of PDAC models with different expression patterns of miR-34a and PLK1 (both high, both low and intermediate levels) which although interesting and important, it is out of the scope of this manuscript.

The concept of RNAi therapy by combining a set of miRNA (520d-3p) and siRNA (EphA2) was also shown to be effective in a recent work in which liposomal nanoparticles were utilized to target oncogenic pathways altered in a different cancer model of ovarian carcinoma[49]. In another study, concurrent delivery of miR-34a and KRAS-siRNA using a lipid/polymer nanoparticle led to antitumor effects in a lung cancer mouse model[50]. The combination treatment of two microRNAs, miR-34a and miR-143/145, entrapped in a lipid-based nanoparticle, has also shown significant therapeutic efficacy in an orthotopic xenograft model of pancreatic cancer[51]. However, combining microRNA mimic with siRNA for pancreatic cancer therapy has not been reported to date.

We showed that these antitumor effects are, at least partially, due to a decrease in MYC, a shared downstream target of both miR-34a and PLK1. Recently, PLK1 was identified as one of MYC-associated proteins using a proteomic approach[52]. PLK1 was also shown to directly interact with MYC and induce MYC phosphorylation[18]. MYC is a central oncogene in PDAC and genetic alterations of MYC were observed in 70% of PDAC patients[53]. MYC, a major regulator of transcription, induces tumor growth, cellular proliferation, protein synthesis, and increased cell metabolism[54]. Because most genetic or epigenetic events in PDAC initiation and progression contribute to MYC activation, its targeting in PDAC is a highly promising therapeutic option. However, development of potent small molecules for MYC inhibition is extremely challenging due to lack of enzymatic activity of MYC and its function through protein–protein and protein–DNA interactions[55]. Therefore, small RNA-based technologies, might serve as a more selective approach to target MYC. As we show here, the combination of miR-34a and PLK1-siRNA downregulated MYC in PDAC cells in vitro and decreased significantly MYC levels in MiaPaCa2 orthotopic tumors in vivo. Moreover, the observation that MYC expression levels in PDAC patients with long-term survival were lower compared to short-term ones strengthens the clinical potential of our combined miRNA–siRNA treatment. Also, the observation that the two patient populations did not differ from each other in the levels of MET and Bcl2 (other targets of miR-34a) emphasizes the specific effect on MYC by our miRNA–siRNA combination treatment. The results showing that the tumorigenic phenotype of MYC-overexpressed-cells could not be rescued by our therapy suggest that the effect of miR-34a and PLK1-siRNA is dependent on MYC; however, we do not rule out the effect of miR-34a in other cancer-associated pathways by targeting additional genes, which might assist in its antitumor phenotype.

Our results demonstrate that small RNA oligonucleotides therapy can synergistically inhibit solid pancreatic tumor growth, and that rationally designed targeted RNA combination nano-therapies carried by our APA nanocarrier may be used to improve therapeutic response in patients who desperately need it.

## Methods

**Survival analysis based on TCGA data**. We downloaded pancreatic cancer patient data from TCGA, which covers 180 patient samples that have available information of gene expression, miRNA expression and clinical data. The samples were divided into 4 groups based on the median expression values of miR-34a and PLK1. We compared the patient survival between high miR-34a/low PLK1 and low miR-34a/high PLK1 (high miR-34a/high PLK1 and low miR-34a/low PLK1 samples were excluded from our survival analysis). We performed Kaplan–Meier analysis to compare patient survival between these two groups using log-rank test. We further controlled for potential confounding factors such as age, sex, race, and metastasis to lymph nodes using a Cox proportional hazard model:

$$h_s(t, \text{patient}) \sim h_{0s}(t)\exp\left(\beta_I I + \beta_{\text{age}}\text{age} + \beta_{\text{lymph}}\text{lymph}\right),$$

where $s$ is an indicator variable overall possible combinations of patients' stratifications based on race and sex. $h_s$ is the hazard function (defined as the risk of death of patients per time unit), and $h_{0s}(t)$ is the baseline-hazard function at time $t$ of the $s^{\text{th}}$ stratification. The model contains three covariates: (i) $I$: indicator variable whether the samples belong to high miR-34a/low PLK1 group or low miR-34a/high PLK1 group, (ii) age: age of the patient, and (iii) lymph: metastasis to lymph nodes. The $\beta$s are the regression coefficients of the covariates, which quantify the effect of covariates on the survival, which were determined by standard likelihood maximization of the model[56].

**FFPE human PDAC specimens**. FFPE specimens were obtained with the approval of the Institutional Review Board (IRB) and in compliance with all legal and ethical considerations for human subject research. A total of 10 FFPE samples were collected from Sheba Medical Center tissue archive: 7 samples of LTS and 3 samples of STS. FFPE samples were analyzed for miRNA levels by real-time qRT-PCR and for PLK1, MYC, Bcl2 and MET levels by immunstaining. Immunostained slides were blindly evaluated by the study pathologist and scored as follows: 0–3: 0- none, 1- weak, 2- moderate, 3- high.

**APA synthesis**. To a solution of PGA (56 mg, 0.43 mmol per monomer) in dry N, N-Dimethylformamide (DMF, 2.5 ml) a solution of Carbonyldiimidazole (86 mg, 0.53 mmol) in dry DMF (1.5 ml) was added. The reaction mixture was stirred for 1.5 h at 25 °C under Argon atmosphere. Tributylamine (0.1 ml, 0.43 mmol) was added and the reaction was left to stir for 5 min. A solution of Hexylamine (19 mg, 0.19 mmol) and Boc-ethylenediamine (42 mg, 0.26 mmol) in dry DMF (2 ml) was added and the reaction mixture was stirred for additional 12 h at the starting conditions. A solution of carbonyldiimidazole (146 mg, 0.9 mmol) in dry DMF (1 ml) was added and the reaction mixture was stirred for additional 12 h at the starting conditions. DMF was removed under reduced pressure. Double distilled water (DDW, 40 ml) was added and the reaction mixture was freeze dried. The resulting solid was dissolved in dichloromethane (DCM, 5 ml) and trifluoroacetic acid (5 ml) was added at 0 °C. The mixture was stirred at 25 °C for 10 min and then evaporated under reduced pressure. The oily residue was dissolved in DDW (40 ml) and the aqueous phase was extracted with DCM (2 × 50 ml) and Diethyl ether (50 ml). The aqueous phase was collected, treated with a 10% NaOH solution to reach pH 5 and freeze dried. The obtained solid was dissolved in DDW (20 ml) and dialyzed for 48 h at 4 °C. The aqueous phase was collected and freeze dried to receive a white powder of trifuoroactice salt, with a 53% yield. For Cy5-labeling of APA, Cyanine dye Cy5-COOH was conjugated directly to the APA backbone. The final Cy5-APA conjugate was purified by dialysis in DDW using a 3.5 kDa a dialysis membrane (GeBaFlex, Gene Bio-Application). Chemical analysis and characterization was published elsewhere[30].

**Fig. 7** In vivo antitumor effect of miRNA–siRNA combination. **a** Trial design for testing miRNA–siRNA combination efficacy in the orthotopic PDAC model. **b** Tumor growth curves from biweekly fluorescent measurements of tumor-bearing mice treated with APA complexed with miR-34a/PLK1-siRNA, miR-34a/NC-siRNA, PLK1-siRNA/NC-miR, NC-miR/NC-siRNA or PBS (treatments are marked with arrows). ($n = 6, 7$). Data represent mean ± SEM. One way ANOVA. **c** In vivo toxicity via mouse body weight evaluation. Data represent mean ± SEM. **d** An image of a representative mouse from each treatment group 33 days post tumor inoculation showing the difference in tumor fluorescent signal. **e** Kaplan–Meier survival graph. Log-Rank test, $P < 0.05$ for the combination miR-34a/PLK1-siRNA compared to all other treatment groups. **f** Effect of miR-34a-PLK1-siRNA combined treatment on proliferation, apoptosis and angiogenesis in MiaPaCa2 orthotopic xenograft tumors. Representative images of H&E, Ki67, cleaved caspase 3 and CD31 immunostaining of tumors from the different treatments, following 45 days from tumor inoculation, are shown (8–10 fields per slide). Scale bars, 200 μm. **g** Quantification of immunostaining that were shown in **f**. MVD, microvessel density. Data represent mean ± SD. (Student's t-test, *$P < 0.05$; **$P < 0.01$; ***$P < 0.001$)

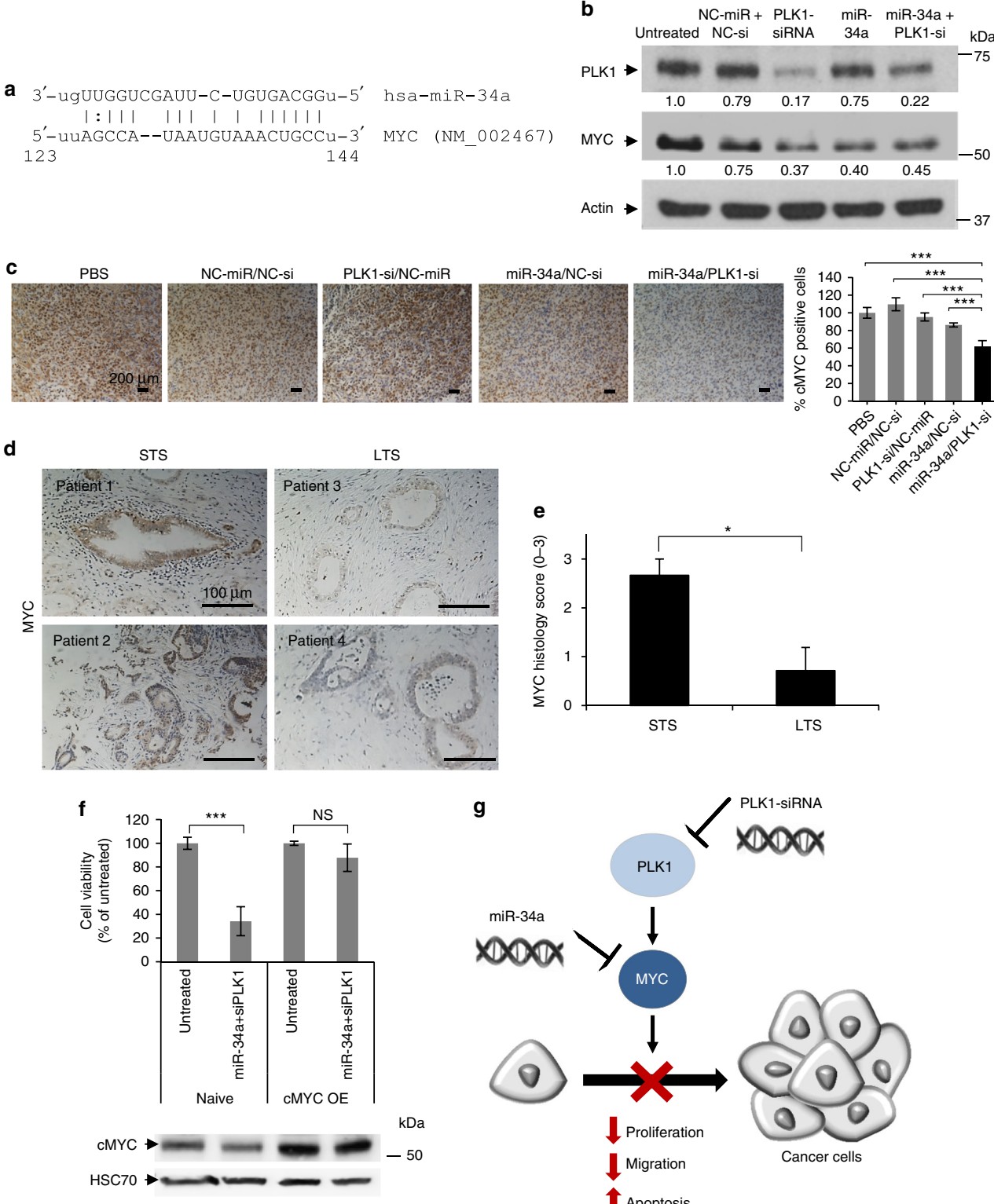

**Fig. 8** Synergistic anticancer effect by the combination of the restoration of miR-34a and silencing of PLK1 is via myc. **a** miR-34a binding site within MYC 3′-UTR. **b** PLK1 and MYC protein levels in MiaPaCa2 cells transfected with miRNA and siRNA monotherapies and their combination. (Representative blot out of 3 biological repeats is shown). **c** MYC immunostaining of tumors from the different treatments of the in vivo experiment shown in Fig. 8. **d** MYC immunostaining of short-term and long-term PDAC FFPE specimens. Representative images are shown. **e** Quantification of MYC immunostaining based on histology scores (0–3: 0- none, 1- weak, 2- moderate, 3- high). **f** Cell viability of cMYC overexpressed-MiaPaCa2 cells (transiently transfected with MYC ORF-containing plasmid 24 h prior to treatments) and naive cells following treatment with the combination for 48 h. Immunoblotting of MYC is depicted beneath the graph (n = 3 biological repeats). **g** Proposed model of synergism via MYC as a common target for miR-34a and PLK1. STS; short-term survivors, LTS; long-term survivors. cMYC OE; cMYC overexpression. Data represent mean ± SD. (Student's t-test, NS; not significant for P > 0.05, *P < 0.05, **P < 0.01)

**Electrophoretic mobility shift assay**. The optimal N/P ratio for polyplex formation was studied using EMSA. miRNA and siRNA (50 pmol total) and increasing amounts of APA were mixed together in RNase-free ultra-pure water (UPW, Biological Industries), incubated at room temperature for 20–30 min and analyzed by electrophoresis on a 2% agarose gel for 30 min at 100 V.

**DLS, zeta potential, and TEM**. Analysis of the mean hydrodynamic diameter of APA-miRNA-siRNA polyplex (0.1 mg APA polymer, N/P 2, in UPW) was performed using VASCO particle size analyzer (Cordouan Technology). Zeta potential of the polyplex (15.5 mM phosphate buffer, pH 7.4) was measured using a Zeta-Sizer Nano ZS instrument with an integrated 4 mW He-Ne laser ($\lambda = 633$ nm; Malvern). All measurements were performed at 25 °C. For TEM: samples were adsorbed on formvar/carbon coated grids and stained with 2% aqueous uranyl acetate. Samples were examined using Jeol 1200EX transmission electron microscope (Jeol).

**Heparin-induced release of miRNA and siRNA**. The relative stability of APA polyplexes was tested by measuring miRNA release from the polyplexes in the presence of a competing polyanion, heparin (Sigma-Aldrich). Polyplex solutions (N/P 2) were incubated in the presence of 0.01–0.1 international units (IU) of heparin 50 pmol$^{-1}$ miRNA/siRNA for 15 min. Samples were analyzed by electrophoresis and run on a 2% agarose gel for 30 min at 100 V.

**Cathepsin B labeling with Cy5-labeled activity-based probe**. PDAC cell lines (300,000 cells per well, 6-well plate) and frozen tissue sections of orthotopic tumor xenografts and normal adjacent tissue were pretreated with either 5 μM cathepsin inhibitor (GB111[34], not fluorescently labeled) or DMSO (0.1%). Then, all samples were incubated for 4 h with Cy5-labeled activity-based probe GB123[33] (0.25 μM for tissues or 2 μM for cells, 0.1% DMSO final concentration). To visualize active cathepsin labeling, tissue sections were stained with DAPI and imaged with a fluorescent microscope equipped with a 630 nm laser. Treated PDAC cells were lysed, separated on 12.5% SDS-PAGE and scanned by Typhoon scanner (GE Healthcare) using an excitation/emission filter set of 635/670 nm. Following scanning, SDS-PAGE was immunoblotted on a PVDF membrane and incubated with beta-Actin antibody (Abcam, 1:1000). Chemiluminescence was measured with a ChemiDocXRS imaging system.

**Cell culture**. Human pancreatic cancer cell lines MiaPaCa2, Panc1, BxPC3 and the murine pancreatic cancer cell line Panc02, were purchased from the American Type Culture Collection (ATCC). KPC cells were established in the laboratory of Surinder K. Batra and derived from pancreatic adenocarcinoma tumors of Kras$^{G12D}$; Trp53$^{R172H}$; Pdx1-Cre (KPC) transgenic mice[57]. KPC cells were provided by Surinder Batra under a signed MTA with The University of Nebraska Medical Center. All cell lines used in this paper are not listed in the database of commonly misidentified cell lines maintained by ICLAC. The cell lines have not been authenticated. All cells were mycoplasma free. Cells were cultured in Dulbecco's modified Eagle's medium (DMEM) supplemented with 10% fetal bovine serum (FBS), 100 U ml$^{-1}$ penicillin, 100 μg ml$^{-1}$ streptomycin and 2 mM L-glutamine (Biological Industries) and grown at 37 °C; 5% $CO_2$.

**Confocal microscopy**. MiaPaCa2 cells ($2 \times 10^5$ per well) were seeded in 24-well plate on cover slips for 24 h. Polyplexes containing APA (3.5 μg ml$^{-1}$) and Cy5-labeled siRNA (100 nM, Syntezza Bioscience) were added to cells and incubated for 4, 24, and 48 h. Slides were washed twice with PBS, fixed in 4% paraformaldehyde (PFA, Electron Microscopy Sciences) at room temperature for 20 min and washed again with PBS. Next, 0.1% Triton X-100 in PBS was added for 2 min and slides were washed twice with PBS. Actin filaments were stained with fluorescein isothiocyanate (FITC)-labeled phalloidin (Sigma-Aldrich) for 40 min and slides were washed and mounted with Prolong Gold antifade reagent with DAPI (Invitrogen). For immunofluorescence staining, fixed permeabilized cells were blocked with normal mouse/rabbit IgG (Santa Cruz Biotechnology, Inc.) in blocking solution (PBS, 2% BSA) for 30 min. Cells were then incubated with mouse anti-EEA1 (BD Biosciences) and rabbit anti-LAMP1 (Cell Signaling) primary antibodies for 1 h and washed three times. Finally, anti-mouse FITC and anti-rabbit Rhodamine (BD Biosciences) secondary antibodies were applied for 30 min at room temperature and slides were washed and mounted as described above. Cellular internalization and trafficking were monitored using Leica SP5 Confocal Imaging system (Leica Microsystems).

**Transfection of APA-miRNA–siRNA polyplex**. PDAC cells were seeded in 6-well plates and 24 h later transfected with polyplexes containing APA and miRNA/siRNA at the indicated concentrations in antibiotic-free medium. Twenty-four or 48 h later, cells were collected, and total RNA or proteins were extracted.

**miRNA/mRNA expression levels determination**. Total RNA from cultured cells was isolated using EZ-RNA II total RNA isolation kit (Biological Industries) according to the manufacturer's instructions. Total RNA from FFPE tissues was isolated using miRNeasy FFPE kit (Qiagen) according to the manufacturer's

instructions. RNA quality was measured using NanoDrop (Thermo Scientific). Reverse transcription reaction for miRNA and mRNA was performed using miScript II Reverse Transcription Kit (Qiagen) with Hiflex buffer supplemented in the kit. miRNA expression levels were quantified by real-time qRT-PCR using miScript Primer assay (Qiagen) and normalized to RNU6. mRNA levels were quantified using custom qRT-PCR primers (Syntezza Bioscience) and normalized to Actin. qRT-PCR for miRNA and mRNAs was performed using miScript SYBR Green PCR kit (Qiagen) or SensiFAST SYBR HiROX kit (Bioline), respectively, according to manufacturer's instructions using StepOnePlus real-time PCR system (Applied Biosystems). Primers were designed as: CDK6 FWD: 5'-TCACGAACAGACA-GAGAAACC-3', CDK6, REV: 5'-CTCCAGGCTCTGGAACTTTATC-3', Notch1 FWD: 5'-GCCTTTGTGCTTCTGTTCTTC-3', Notch1 REV: 5'-CTGGCCTCA-GACACTTTGAA-3', Bcl2 FWD: 5'-GGCCAGGGTCAGAGTTAAATAG-3', Bcl2 REV: 5'-GGAGGTTCTCAGATGTTCTTCTC-3', MET FWD: 5'-CAGTGGTGG-GAGCACAATAA-3', MET REV: 5'-TGTAAAGTTCCTTCCTGCTTCA-3'. PLK1 FWD: 5'-CACAGTTTCGAGGTGGATGT-3', PLK1 REV: 5'-ATCCGGAGG-TAGGTCTCTTT-3'. Expression values were calculated based on the comparative threshold cycle (Ct) method.

**Western blot analysis**. Cells were homogenized with lysis buffer and debris was removed by centrifugation. Protein concentration was determined using the BCA protein assay kit (Thermo Scientific). Lysates were loaded on 10% SDS–PAGE gel and transferred by electroporation to nitrocellulose membrane. Membranes were blocked for 1 h in TBST buffer containing 5% milk and incubated with mouse anti-cMET (cat# 3127, 1:1000, Cell Signaling Technology), mouse anti-CDK6 (cat# 3136, 1:2000, Cell Signaling Technology), rabbit anti-Bcl2 (cat# 4223, 1:1000, Cell Signaling Technology), rabbit anti-Notch1 (cat# 4380, 1:1000, Cell Signaling Technology), rabbit anti-PLK1 (cat# 4535S, 1:500, Cell Signaling Technology) or rabbit anti-MYC (cat# 10828-1-AP, 1:1000, Proteintech). Mouse anti-HSC70 (cat# SC-7298, 1:500, Santa Cruz Biotechnology, Inc.) or goat anti-actin (cat# SC-1616, 1:500, Santa Cruz Biotechnology, Inc.) were used for loading control. Analysis was performed using HRP-conjugated secondary antibodies: rabbit anti-goat (cat# AP106P, 1:1000, Chemicon International), goat anti-mouse (cat# ab7068, 1:10,000, Abcam) and goat anti-rabbit (cat# SC-2004, 1:10,000, Santa Cruz) followed by ECL-PLUS Detection Kit (Pierce). Band quantification was performed using ImageJ software (National Institutes of Health) and protein levels were normalized to Actin or HSC70 levels. For uncropped scans of blots see Supplementary Fig. 13.

**Cell viability assay**. MiaPaCa2, BxPC3 ($2 \times 10^4$ cells per well), KPC and Panc1 ($1 \times 10^4$ cells per well$^{-1}$) were plated onto 24-well plates. Twenty-four hours later, cells were transfected with APA-miRNA–siRNA polyplexes. Following 72 h incubation, cells were washed, detached by trypsin and viable cells were counted by Z1 Coulter Counter (Beckman Coulter).

**Live cell uptake of APA-siRNA polyplexes**. Internalization of Cy5-labeled siRNA into live PDAC cells was followed using ImageStream 100 Imaging Flow Cytometer (Amnis). Live cells ($2 \times 10^6$ 50 μl$^{-1}$ PBS supplemented with 0.5% BSA) were monitored 24 h following transfection with Cy5-labeled siRNA (100 nM) alone or complexed with APA. Lipofectamine 2000 (ThermoFisher Scientific) was used according to the manufacturer's instructions and served as control for transfection.

**Cell migration**. To study the ability of APA-miR-34a-PLK1-siRNA polyplex to inhibit the migration of PDAC cells, we used the IncuCyte ZOOM Live Cell Imaging system (Essen BioScience). Cells were plated onto 96-well ImageLock tissue culture plate (60,000 MiaPaCa2 and 30,000 KPC cells per well) in DMEM supplemented with 10% FBS, 2 mM L-glutamine and were allowed to grow until confluency (37 °C; 5% $CO_2$). Then, treatments of APA-miRNA–siRNA (100 nM miRNA, 50 nm siRNA) polyplexes were added to the cells for 20 h. Next, a wound was created in each well using a 96-pin wound-making tool (WoundMaker, Essen BioScience), dislodged cells were washed with DMEM medium and treatments were re-added. To monitor wound closer, the plate was placed in the IncuCyte incubator and phase contrast images were taken at regular intervals over a course of 48 h using ×10 objective. Results were calculated by the IncuCyte Software and presented as wound confluence relative to time 0 or as relative wound density (which accounts for changes in cell densities inside and outside the wound region relative to the background density of the wound at time 0).

**Colony formation**. PDAC cells were transfected with APA (3.5 μg ml$^{-1}$) polyplexes containing miR-34a/non targeted negative control (NC)-miR/ PLK1-siRNA/non targeted negative control (NC)-siRNA (100 nM) alone or combined, or left untreated, for 24 h. Transfected cells were reseeded in 35 mm plates (250 cells per plate for KPC, 500 cells per plate for MiaPaCa2 and 1000 cells per plate for Panc1 and BxPC3) in triplicates. Cells were allowed to grow for 8–14 days, then, plates were washed with PBS, stained with crystal violet for 30 min, washed with DDW and photographed. Quantification of colonies area was done using ImageJ software and presented as total area.

**Human PBMCs isolation and cytokine ELISA.** Fresh human peripheral blood mononuclear cells (PBMCs) were isolated from leukocyte enriched whole blood of healthy donors obtained from Sheba Medical Center Blood Bank. Informed consent was obtained from all blood donors used in these studies. Whole blood was diluted with RPMI 1640 (1:1 ratio). The diluted blood was gently overlaid onto 10 ml Ficoll (GE Healthcare) at 1:2 ratio. Gradients were centrifuged at 22 °C, 900×$g$, for 25 min. Opaque-light PBMCs ring was transferred into a new tube, washed with RPMI and centrifuged at 800×$g$ for 10 min. PBMCs ($3 \times 10^6$) were seeded in 12-well plates (1 ml per sample) and cultured in triplicates in RPMI 1640 medium supplemented with 10% FBS, 2 mM L-Glutamine, 1 mM pyruvate, 1% non-essential amino acids, 100 U ml$^{-1}$ penicillin and 100 μg ml$^{-1}$ streptomycin. Treatments of APA-siRNA (50 and 200 nM) polyplexes were added to PBMCs. APA alone was used as vehicle control. PBMCs growth medium served as negative control. Lipopolysaccharides (LPS, 2 μg ml$^{-1}$, Sigma-Aldrich) served as positive control. Culture supernatants were collected after 24 h and assayed for human interleukin 6 (IL-6) and tumor necrosis factor alpha (TNF-α) cytokines by sandwich ELISA kits (R&D Systems).

**Hemolysis assay.** Rat red blood cells (RBC) solution (2% wt/wt) was added to a 96-well plate and incubated with serial dilutions (1–10,000 μg ml$^{-1}$) of APA: miRNA polyplexes (N/P 2) for 1 h at 37 °C. The highest polyplex concentration used in this assay (10,000 μg ml$^{-1}$), was 100-fold higher than the one used in the in vivo experiment (100 μg ml$^{-1}$ polymer for 3 mg kg$^{-1}$ oligonucleotide dose in 2 ml mouse blood volume). Following plate centrifugation, the supernatants were transferred to a new plate and absorbance was measured at 550 nm using a SpectraMax M5e plate reader (Molecular Devices). Sodium dodecyl sulfate (SDS) was used as positive control, whereas dextran (Mw 70 kDa, Sigma-Aldrich) was used as negative control.

**Polyplex stability in serum.** To determine the ability of APA nanocarrier to stabilize miRNA against serum degradation, APA-miR-34a polyplexes were incubated with 100% FBS for 12 h at 37 °C and run on 2% agarose gel. miRNA bands were digitized and quantified using ImageJ software to determine the mean density of the miRNA bands.

**Generation of mCherry-labeled human PDAC MiaPaCa2 cell line.** mCherry-labeled MiaPaCa2 cell line was generated as previously described[58]. Briefly, human embryonic kidney 293T (HEK293T) cells were co-transfected with pQC-mCherry and the compatible packaging plasmids, pMD.G.VSVG and pGag.pol.gpt. Forty-eight h following transfection, pQC-mCherry retroviral particles-containing supernatants were collected and filtered (0.45 μm). Cells were infected with the retroviral particles-containing media and mCherry positive cells were selected 48 h following the infection by puromycin (9 μg ml$^{-1}$) resistance.

**Animal studies.** All animals were housed in Tel Aviv University's animal facility and the experiments were approved by our institutional animal care and use committee (IACUC) and conducted in accordance with NIH guidelines. For orthotopic cell inoculation, severe combined immunodeficiency (SCID) male mice (Envigo) aged 5 weeks were anesthetized using ketamine (100 mg kg$^{-1}$) and xylazine (12 mg kg$^{-1}$) and a subcostal left incision of 1 cm in the skin and in the peritoneum was done. The pancreas was exposed and $1 \times 10^6$ mCherry-labeled MiaPaCa2 cells suspended in 20 μl PBS were injected into the pancreas using a Hamilton syringe with a 30 G needle. The incision was closed with a 5–0 nylon monofilament suture. At the end of the surgery, mice received Rimadyl (100 μl per 25 g body weight) for pain relief and were monitored until recovery. Tumor growth was monitored twice a week using an intravital non-invasive fluorescence imaging system (Maestro, Cambridge Research and Instrumentation). For maximum tolerated dose (MTD) experiment, Balb/c mice, body weight- 25 g, were injected with increasing amounts of APA-miRNA–siRNA polyplexes dosages and monitored for general health in the 5 following weeks. To assess the polyplexes effect on mouse normal pancreas and on glucose levels in mouse blood, 5-week-old male C57BL/6 mice were injected with APA-miRNA–siRNA (2 mg kg$^{-1}$ oligonucleotide dose) polyplexes 3 sequential injections. At various time points, blood glucose levels were measured from the mice tail by a glucometer. The mice were anesthetized and pancreas was resected, embedded in paraffin and stained with Hematoxylin and Eosin. For antitumor activity of APA-miRNA–siRNA polyplexes, mice bearing orthotopic tumors were randomized and divided into 5 treatment groups, each group with an average mCherry tumor signal of ~100 scaled counts per s: (i) PBS ($n = 7$), (ii) APA-miR-34a/PLK1-siRNA ($n = 7$), (iii) APA-miR-34a/NC-siRNA ($n = 6$), (iv) APA-PLK1-siRNA/NC-miR ($n = 6$) and (v) APA-NC-miR/NC-siRNA ($n = 6$). miR-34a/NC-miR were from Biospring and PLK1-siRNA/NC-siRNA were from Syntezza Bioscience. The mice were injected intravenously five consecutive times, for two cycles with a 3 days break between them with total of 3 mg kg$^{-1}$ oligonucleotide dose per treatment. In all animal studies, mice were monitored twice a week for general health, body weight and tumor signal in a non-blinded manner. At day 40 from tumor inoculation, blood was withdrawn from mice and analyzed for blood count and chemistry (American Medical Laboratories).

**mCherry-labeled tumor-bearing mice imaging.** Tumor formation and progression was monitored twice a week by a non-invasive fluorescence imaging system

(CRI-Maestro). The mice were anesthetized, treated with depilatory cream and placed inside the imaging system. Multispectral image-cubes were acquired through 550–800 nm spectral range in 10 nm steps using excitation (595 nm long pass) and emission (645 nm long pass) filter set. The mice auto-fluorescence and undesired background signals were eliminated by spectral analysis and linear unmixing algorithm of the CRI-Maestro software.

**APA:Cy5-labeled siRNA polyplex biodistribution.** For co-localization: mice bearing orthotopic mCherry-labeled MiaPaCa2 tumors with a fluorescent signal of ~1000 scaled counts per second were administered via the tail vein with APA: Cy5-labeled siRNA polyplexes (N/P 2, 0.5 mg kg$^{-1}$ siRNA, 100 μl). Mice were imaged over time and Cy5 fluorescent signal from whole body was measured using the CRI-Maestro imaging system. Co-localization of mCherry and Cy5 was determined by the CRI-Maestro software. For biodistribution: tumors and healthy organs (heart, lungs, liver, kidneys, and spleen) were resected from mice 24 h following i.v. injection of either the APA:Cy5-labeled siRNA polyplex or Cy5-labeled siRNA alone and fluorescent intensity of each organ was measured in the CRI-Maestro.

**Immunohistochemistry.** FFPE samples of tumors were sectioned at 5 μm thickness, mounted on positively charged glass slides and dried for 30 min at 37 °C. One section per sample was deparaffinized, rehydrated, and stained with hematoxylin and eosin (H&E). Additional sections were immunostained using the automated immunohistochemistry (IHC) and in situ hybridization staining system Bond RX (Leica Biosystems). First, sections were submitted to heat-induced epitope retrieval with Epitope Retrieval solution 1 (ER1, AR9961, Leica Biosystems) for 20 min. Endogenous peroxidase activity was blocked with 3–4% (v/v) hydrogen peroxide (part of DS9263, Leica Biosystems) for 12 min. Afterward, sections were incubated with goat blocking serum (Biological Industries) for 35 min. Then, staining was performed using Intense R Detection system (Leica Biosystems) according to the manufacturer's protocol. Primary antibody incubation time was 1 h. Sections were stained for proliferating cells using rabbit anti- Ki67 antibody (1:200, cat# RBK027–05, Zytomed), apoptotic cells using rabbit anti-cleaved caspase 3 antibody (1:100; cat# CP229A, Biocare Medical) and microvessels using rat anti- CD31 antibody (1:20, cat# DIA310, Dianova). Sections were also stained with rabbit anti-PLK1 (1:50, cat# bs-3535R, Bioss Antibodies), rabbit anti-MYC (1:100, cat# 10828-1-AP, Proteintech) and mouse anti-αSMA (1:300, cat# A2547, Sigma-Aldrich). Microvessel density (MVD) was calculated as previously described[59]. Briefly, tumors were scanned for areas of high vessel density (i.e., "hot spots") at low power. Then, individual microvessels were counted at a higher power (×400 field). Vessels with an open lumen that, in most cases, contained red blood cells were defined as vessels positively stained for CD31, and counted at high power (×400 field). Staining intensities of Ki67, cleaved caspase 3, PLK1 and MYC from tumor sections were determined using ImageJ software with Fiji plugin[60] (downloaded from http://fiji.sc).

**Vasculature/stroma morphologies of PDAC xenograft and normal murine pancreas.** To monitor PDAC xenograft and normal pancreas vasculatures, we administered i.v. FITC-labeled dextran (70 KDa, 10 mg ml$^{-1}$, 200 μl) to orthotopic tumor-bearing mice. Then, the pancreas was exposed and vasculature functionality and morphology were monitored by a fiber confocal microscopy imaging system (CellVizio, Mauna Kea Technologies). Mean Vessel Diameter (MVD) of blood vessels within PDAC tumor and normal pancreas and fluorescent signal in areas outside the vessels were measured using the CellVizio imaging Software (Mauna Kea Technologies). Blood vessels and activated fibroblasts in PDAC tumor xenograft and normal pancreas were immunostained using anti-CD31 and anti-αSMA antibodies, respectively (see IHC section).

**Statistical methods.** Data are expressed as mean ± standard deviation (SD) for in vitro assays and as mean ± standard error of the mean (SEM) for in vivo assays. Statistical analyses were performed with Student's $t$-test unless noted otherwise. Statistical significance in mice OS was determined by the log-rank test using SigmaPlot software (Systat Software Inc.). The $P$-values are *$P < 0.05$, **$P < 0.01$ and ***$P < 0.001$. $P < 0.05$ was considered statistically significant.

**Data availability.** The open-access TCGA data that support the findings of this study are available from https://gdc-portal.nci.nih.gov/. The authors declare that the data supporting the findings of this study are available within the paper and its Supplementary Information. Additional data and source files are available from the corresponding author upon reasonable request.

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

## Acknowledgements

The Satchi-Fainaro laboratory's research leading to these results has received partial funding from the European Research Council (ERC) under the European Union's Seventh Framework Programme/ERC Consolidator Grant Agreement n. [617445]-PolyDorm, THE ISRAEL SCIENCE FOUNDATION (Grant No. 918/14), the framework of Rimonim Consortium and the MAGNET Program of the Office of the Chief Scientist of the Israel Ministry of Industry, Trade and Labor (54206), and Tel Aviv University Cancer Biology Research Center (CBRC) (Grant No. 2040). H.G. thanks the Marian Gertner Institute for scholarship awarded to outstanding students doing research in the field of Medical Nanosystems at Tel Aviv University. We thank Y. Zilberstein (Tel Aviv

University) for intensive support in the animal imaging facility. We thank L. Mittelman and S. Lichtenstein (Tel Aviv University) for excellent assistance with the confocal imaging. We thank E. Yeini and M. Goldenfeld (Tel Aviv University) for constructive help with the Bond RX system and the IHC staining. We thank Prof. Moshe Oren (The Weizmann Institute of Science) for providing us the cMYC ORF plasmid.

## Author contributions

H.G. performed the in vitro studies; H.G. and D.B.-S. performed the in vivo studies; S.E., A. K., and R.B. synthesized nanocarriers and performed physicochemical characterization; H. G. and G.T. performed the immunohistochemistry experiments; Y.E., R.B., E.M., and G.B. designed and performed cathepsin assays; H.G., S.E., P.O., L.L., and R.S.-F. were involved in study conception, experimental design and analysis of the data; J.S.L. and E.R. analyzed TCGA data. I.B. and T.G. analyzed and interpreted the pathology data. H.G. and R.S.-F wrote and reviewed the manuscript. All authors discussed the results and commented on the manuscript.

## Additional information

**Competing interests:** The authors declare no competing financial interests.

