## [Peer Review File · Nature Communications]

Reviewers' comments:

Reviewer #1 (Remarks to the Author):

The authors have shown a sufficiently proven hypothesis to combine siRNA and miRNA34a for targeted killing of PDAC cells with the aid of stabilisation using polymeric nanocarrier.

The manuscript is well structured and easy for readers to follow.

There are some questions that maybe authors can address.

1. Figure 1D. does the miR34a expression mean the RQ from the qRT-PCR? Figure 1D looks ok but could probably benefit if the PLK1 expression level can be also shown from IHC or Western data to easily show negative correlation.

2. Figure 2b. The authors showed when Nitrogen/phosphate ratio at 2 the siRNA and miRNA are loaded with nanocarriers. However, do you expect or have evidence that the loading is equal? (although figure 2e shows release of miRNA from nanocarrier from heparin treatment.)

3. For the total area in comparison of control in colony formation assay in figure 4jk and Supplementary figure 4. Are the data shown correctly with error bar as SD very small? I just wonder if the error bars are so small yet the t-test shown non significant between control and mock siRNA group.

minor typos:

1. pg. 3 line25, the comma between 16 and 17 should be upper script.
2. pg.24 line 24, MYC, a shared downstream target.
3. pg.27 2mM L-glutamine

Reviewer #2 (Remarks to the Author):

In this study, Gibori describe a novel delivery method for miRNA/siRNA based therapeutics to tumor cells. They identify two targets (miR-34a and PLK1) through interrogation of TCGA gene expression and survival data and utilize an APA-based nanocarrier to deliver them to orthotopic pancreatic tumor xenografts. The study is well written and clearly describes the authors' systematic approach to testing their new delivery system. They show that their therapy decreased the proliferation and migration of a single human pancreatic cancer cell line, both in vitro and in vivo, and they propose a molecular mechanism through which this might happen. The proposed therapeutic approach is quite novel and thus likely to be of great interest to others in the field. However, the study could be approved with attention to the following:

1. The major weakness of the study is that the vast majority of experiments are performed in only a single cell line (MiaPaCa2). While the authors very convincingly show that their approach can treat MiaPaCa2 tumors, it is not clear how broadly effective their therapy will be. They do show in vitro efficacy against murine KPC lines, but it is unclear how applicable this result would be to human tumors. Would it be possible to show efficacy in xenografts with another human cell line? Can they provide data or cite specific papers from the literature that demonstrate high expression of cathepsin in human PDACs to support that their delivery method would work in human tumors? This reviewer recognizes that repetition of the entire study with a different cell line is not a reasonable request for this publication, but at the very least the authors should acknowledge this caveat in the Discussion. What evidence can they provide that this therapy will be effective beyond MiaPaCa2?

2. In the first section of the results, the authors describe two groups of patients - high miR-34a/low PLK1 and low miR-34a/high PLK1. Were there any patients in which miR-34a and PLK1 were either both high or both low? Or did were the levels always inversely correlated? If the latter, is there a possible explanation? A deeper discussion of the correlation of these two biomarkers would be helpful - it is mentioned briefly for the validation data but not in the original analysis of TCGA data.

3. The findings regarding MYC are likely a bit overstated. The authors have demonstrated the effect of their therapy on MYC, but they have not shown that MYC is the mediator of the anti-tumor properties of their therapy. This is a reasonable hypothesis, but further experiments are necessary to demonstrate this.

4. The authors mention two groups of patients based on expression of miR-34a and PLK1. Can they speculate as to which group of patients would be best suited for their therapy? Would it only work on patients who are miR-34a low/PLK1 high? They should discuss these implications for moving their therapy forward.

Reviewers' comments:

Reviewer #1 (Remarks to the Author):

The authors have shown a sufficiently proven hypothesis to combine siRNA and miRNA34a for targeted killing of PDAC cells with the aid of stabilisation using polymeric nanocarrier.

The manuscript is well structured and easy for readers to follow.

We thank the reviewer for the supportive comment.

There are some questions that maybe authors can address.

1. Figure 1D. does the miR34a expression mean the RQ from the qRT-PCR ?

Yes. It is the RQ (average RQ of LTS group and average RQ of STS group) from the qRT-PCR results shown in Figure 1b. The title of the Y axis was corrected accordingly.

Figure 1D looks ok but could probably benefit if the PLK1 expression level can be also shown from IHC or Western data to easily show negative correlation.

We modified the graph shown in Figure 1d to demonstrate in parallel the quantification of miR-34a as an average RQ of Figure 1b versus the quantification of PLK1 staining as quantified by an experienced blinded pathologist co-author on this paper (Dr Iris Barshack, Head of Pathology, Sheba Medical Center) while dividing the groups to LTS and STS (X axis) showing the inverse correlation. The levels of PLK1 were scored and defined in the scale of 0 - 3, where 0 - none, 1- weak, 2 - moderate, 3 - high.

2. Figure 2b. The authors showed when Nitrogen/phosphate ratio at 2 the siRNA and miRNA are loaded with nanocarriers. However, do you expect or have evidence that the loading is equal? (although figure 2e shows release of miRNA from nanocarrier from heparin treatment).

At Nitrogen/phosphate (N/P) ratio of 2, where the complexation between the polymer and the oligonucleotides is optimal, the ratio between the siRNA and miRNA is *circa* 1:1. This is evidenced by two experiments. The first is shown in Figure 2b which is quoted by the reviewer where it can be seen that at this N/P ratio of 2 (APA / RNAi), there is no free RNAi (not PLK1 siRNA nor miR-34a). The Second, we have shown by heparin displacement assay in Figure 2e that the miR-34a is completely displaced by 0.06 U of heparin. Now, we have added a similar assay for the PLK1 siRNA where it can be seen that 0.05 U of heparin displace the PLK1 siRNA.

**Heparin displacement assay:
SE36.7:Plk1 siRNA**

siRNA 0.01 0.02 0.03 0.04 0.05 0.06 0.07 0.08 0.09 0.1 Heparin U

3. For the total area in comparison of control in colony formation assay in figure 4jk and Supplementary figure 4. Are the data shown correctly with error bar as SD very small? I just wonder if the error bars are so small yet the t-test shown non significant between control and mock siRNA group.

We thank the reviewer for the careful note. Indeed, we had a mistake in data of the groups taken for the calculation of the SD. Fig. 4k and Supplementary fig. 4 are now corrected.

minor typos:

1. pg. 3 line25, the comma between 16 and 17 should be upper script

corrected

2. pg.24 line 24, MYC, a shared downstream target.

corrected

3. pg.27 2mM L-glutamine

corrected

Reviewer #2:

In this study, Gibori describe a novel delivery method for miRNA/siRNA based therapeutics to tumor cells. They identify two targets (miR-34a and PLK1) through interrogation of TCGA gene expression and survival data and utilize an APA-based nanocarrier to deliver then to orthotopic pancreatic tumor xenografts. The study is well written and clearly describes the authors' systematic approach to testing their new delivery system. They show that their therapy decreased the proliferation and migration of a single human pancreatic cancer cell line, both in vitro and in vivo, and they propose a molecular mechanism through which this might happen. The proposed therapeutic approach is quite novel and thus likely to be of great interest to others in the field.

We thank the reviewer for the supportive comment.

However, the study could be approved with attention to the following:

1. The major weakness of the study is that the vast majority of experiments are performed in only a single cell line (MiaPaCa2). While the authors very convincingly show that their approach can treat MiaPaCa2 tumors, it is not clear how broadly effective their therapy will be. They do show in vitro efficacy against murine KPC lines, but it is unclear how applicable this result would be to human tumors. Would be it be possible to show efficacy in xenografts with another human cell line? Can they provide data or cite specific papers from the literature that demonstrate high expression of cathepsin in human PDACs to support that their delivery method would work in human tumors? This reviewer recognizes that repetition of the entire study with a different cell line is not a reasonable request for this publication, but at the very least the authors should

acknowledge this caveat in the Discussion. What evidence can they provide that this therapy will be effective beyond MiaPaCa2 ?

As our delivery system, APA, is biodegradable by cathepsin B, we verified that the 2 cell lines used in the study, the murine KPC and the human MiaPaCa2, express the enzyme. Furthermore, we relied on vast data from the literature demonstrating high expression of cathepsin in human PDACs, for example:

1. Dumartin L *et al.*, AGR2 is a novel surface antigen that promotes the dissemination of pancreatic cancer cells through regulation of cathepsins B and D. *Cancer Research* 71(22):7091-102 (2011).
2. Eser S *et al.*, In vivo diagnosis of murine pancreatic intraepithelial neoplasia and early-stage pancreatic cancer by molecular imaging, *PNAS* 108(24):9945-50 (2011).
3. Cruz-Monserrate Z *et al.*, Detection of pancreatic cancer tumours and precursor lesions by cathepsin E activity in mouse models, *Gut* 61(9):1315-22 (2011).
4. Singh N, Saraya A, Roles of cathepsins in pancreatic cancer, *Tropical Gastroenterology* 37(2):77-85 (2016).

In order to experimentally address the reviewer's comment, we tested 5 different pancreatic tumors and found that all (but one) express cathepsin B- see results in a gel form below showing cathepsin B activity using a fluorescent activity-based probe developed by one of the co-authors (Blum G *et al.*, *Nat Chem Biol.* 2007 Oct;3(10):668-77). Taken together, our findings and the published data suggest that indeed most pancreatic cancers express cathepsin B, hence a cathepsin-B-degradable system is relevant for RNAi delivery for PDAC.

Evidence that this therapy will be effective beyond MiaPaCa2:

We would like to note that all the *in vitro* data was replicated in cell line originated from the GEMM KPC mouse model which was shown to be of relevance to the human clinical set-up. KPC mice conditionally express endogenous mutant KRAs and p53 alleles in pancreatic cells (refs 1,2 below), which develop pancreatic tumors whose pathophysiological and molecular features resemble those of human PDA. Furthermore, Olive *et al.* had used the KPC mice to investigate why PDA is insensitive to chemotherapy (ref 3).

1. Olive KP, Tuveson DA, The Use of Targeted Mouse Models for Preclinical Testing of Novel Cancer Therapeutics, *Clin Cancer Res.* 2006 Sep 15;12(18):5277-87.

- Hingorani SR *et al.*, Trp53R172H and KrasG12D cooperate to promote chromosomal instability and widely metastatic pancreatic ductal adenocarcinoma in mice, *Cancer Cell* 7(5):469-83 (2005).
- Olive KP *et al.*, Inhibition of Hedgehog Signaling Enhances Delivery of Chemotherapy in a Mouse Model of Pancreatic Cancer, *Science* 324(5933):1457-61 (2009).

Still, to address the reviewer's concern, similar to the cathepsin B assay that we added above, we tested the ability of our siPLK1-miR-34a APA treatment to inhibit the proliferation and colony formation of two additional human PDAC cell lines, BxPC3 and PANC1. As it can be seen below, we found the inhibition to be synergistic as was shown with the previous two cell lines, MiaPaCa2 and KPC (we added it to the Supplementary, Fig. 7 A,B,E- Panc1 and C,D,F- BxPC3).

We also found supportive evidence in the literature that this therapy will be effective beyond MiaPaCa2: A paper showing that the expression of miR-34a is frequently lost in 15 pancreatic cancer cell lines compared to normal pancreatic cells (HPNE and HPDE) (Chang TC *et al.*, Transactivation of miR-34a by p53 broadly Influences Gene Expression and Promotes Apoptosis, *Molecular Cell*, 26(5):745-52, 2007). See figure below.

Figure 3.
Expression of miR-34a Is Frequently Lost in Pancreatic Cancer Cells
 Northern blot analysis of miR-34a expression in nontransformed pancreatic ductal epithelial cell lines (HPNE and HPDE, in bold) and in pancreatic cancer cell lines. Relative expression of miR-34a in each cell line, normalized to U6 snRNA expression, is shown below the blots. Dashes indicate undetectable miR-34a expression.

Regarding PLK1 overexpression in PDAC, it was shown that invasive pancreatic adenocarcinomas were PLK1 positive in 47.7% of cases (out of 86 patients) (Weichert W *et al.*, Overexpression of Polo-like kinase 1 is a common and early event in pancreatic cancer. *Pancreatology*, 5(2-3):259-65, 2005).

Another article (Song B *et al.*, Plk1 Phosphorylation of Orc2 and Hbo1 Contributes to Gemcitabine Resistance in Pancreatic Cancer, *Molecular Cancer Therapeutics*, 12(1):58-68, 2013) showed IHC staining of 140 patient samples and 80% were positive for PLK1:

The data describing these results were added to the manuscript.

2. In the first section of the results, the authors describe two groups of patients - high miR-34a/low PLK1 and low miR-34a/high PLK1. Were there any patients in which miR-34a and PLK1 were either both high or both low? Or did were the levels always inversely correlated? If the latter, is there a possible explanation? A deeper discussion of the correlation of these two biomarkers would be helpful - it is mentioned briefly for the validation data but not in the original analysis of TCGA data.

As described in the caption of Figure 1 and Methods, we simply divided the patients into 4 groups based on the median values of miR-34a and PLK1, and compared the patient survival between high miR-34a/low PLK1 and low miR-34a/high PLK1. As the reviewer noted, there exist high miR-34a/high PLK1 and low miR-34a/low PLK1 samples, but they were excluded from our survival analysis to focus on the feasibility of our intervention to impact patient survival. In fact, we observe miR-34a and PLK1 expression is marginally positively correlated (Spearman R=0.19, P<0.01, as shown below), and our intervention is likely to revert this correlation, bring the tumor to a state of lower fitness, that leads to improved patient survival.

We explicitly noted this information in the Methods section.

Figure R1. Correlation between PLK1 (X-axis) and miR-34a (Y-axis) expression in TCGA PDAC samples (N=180).

3. The findings regarding MYC are likely a bit overstated. The authors have demonstrated the effect of their therapy on MYC, but they have not shown that MYC is the mediator of the anti-tumor properties of their therapy. This is a reasonable hypothesis, but further experiments are necessary to demonstrate this.

To address the reviewer's comment, we added an experiment and yet, we also understated our findings and allowed for other molecular contributors/pathways to the synergistic effect beside MYC. Therefore, we first set to use a strategy of myc overexpression system, which was kindly provided to us by Moshe Oren from the Weizmann Institute of Science.

We evaluated the ability of our siPLK1-miR-34a APA treatment to rescue the effect resulting from the MYC overexpression.

These results were added to the manuscript (new Fig. 8f).

We added the following to the main text:

To further verify our hypothesis that MYC is the mediator of the anti-tumor properties of our therapy, we evaluated whether overexpression of MYC could rescue the tumorigenic phenotype of PDAC cells. MiaPaCa2 cells were transiently transfected with MYC-ORF plasmid that lacks the 3'-UTR and thus miR-34a binding site. Following MYC overexpression which was confirmed by immunoblotting (Fig. 8f), cells were treated with miR-34a PLK1-siRNA combination or left untreated. Cell viability of naïve (Wild Type) MiaPaCa2 PDAC cells was decreased when treated with the combination. However, MYC-overexpressed cells treated with the combination showed no significant difference in cell viability compared to MYC-overexpressed-cells that were not treated (Fig.8f). These results suggest that the effect of the combination of miR-34a and PLK1-siRNA is dependent on MYC downregulation.

We added the following to the Discussion:

These results suggest that the effect of miR-34a and PLK1-siRNA is dependent on MYC, however, we do not rule out the effect of miR-34a in other cancer-associated pathways by targeting additional genes, which might assist in its antitumor phenotype.

In addition, we used a commercial MYC inhibitor (Sigma) to determine whether there is any additional anti-proliferative effect when adding our treatment. We observed that when treated with the MYC inhibitor at a concentration of 60 μ M, MiaPaCa2 cells's viability decreased to 45%. However, when adding the combination treatment to the MYC inhibitor, there was further decrease in cell viability to 24% as well as decreased expression of MYC protein. This further suggests that our miR-siRNA treatment is dependent on MYC downregulation.

4. The authors mention two groups of patients based on expression of miR-34a and PLK1. Can they speculate as to which group of patients would be best suited for their therapy? Would it only work on patients who are miR-34a low/PLK1 high? They should discuss these implications for moving their therapy forward.

We thank the reviewer for the interesting comment and we clarified this point in the Discussion section as this is only our speculation. Based on the TCGA data and our results on several human and murine cell lines, we can determine that our siPLK1-miR-34a APA treatment is clearly relevant to patients with inverse correlation expression of low miR-34a and high PLK1. However, as high and low levels are relative to an unknown control in the clinic, it seems to us that in the case of PDAC, upregulating miR-34a and

downregulating PLK1 will always be beneficial even if both markers are low or both are high. To prove this, we will need to evaluate our system on an array of PDAC models with different expression patterns of miR-34a and PLK1 (both high, both low and intermediate levels) which although interesting and important, it is obviously out of the scope of this manuscript.

We added this to the Discussion.

REVIEWERS' COMMENTS:

Reviewer #1 (Remarks to the Author):

The authors have addressed all the concerns i made in current revision. The novelty of the carrier-bearing miRNA-siRNA combinatorial therpeutic development and sufficient animal experiment evidence described by the authors would be interesting to the audience of nature communication.

Reviewer #2 (Remarks to the Author):

The authors have very effectively addressed the concerns raised in the first round of peer review. Their additional experiments have further demonstrated the potential clinical applicability of this novel therapy.